# Engineered bioorthogonal POLY-PROTAC nanoparticles for tumour-specific protein degradation and precise cancer therapy

Jing Gao[1,2], Bo Hou[2,3], Qiwen Zhu[2], Lei Yang[4], Xingyu Jiang[4], Zhifeng Zou[2,3], Xutong Li[2], Tianfeng Xu[2], Mingyue Zheng [2], Yi-Hung Chen [5], Zhiai Xu[3] ✉, Huixiong Xu [6] ✉ & Haijun Yu [2,4] ✉

PROteolysis TArgeting Chimeras (PROTACs) has been exploited to degrade putative protein targets. However, the antitumor performance of PROTACs is impaired by their insufficient tumour distribution. Herein, we present de novo designed polymeric PROTAC (POLY-PROTAC) nanotherapeutics for tumour-specific protein degradation. The POLY-PROTACs are engineered by covalently grafting small molecular PROTACs onto the backbone of an amphiphilic diblock copolymer via the disulfide bonds. The POLY-PROTACs self-assemble into micellar nanoparticles and sequentially respond to extracellular matrix metalloproteinase-2, intracellular acidic and reductive tumour microenvironment. The POLY-PROTAC NPs are further functionalized with azide groups for bioorthogonal click reaction-amplified PROTAC delivery to the tumour tissue. For proof-of-concept, we demonstrate that tumour-specific BRD4 degradation with the bioorthogonal POLY-PROTAC nanoplatform combine with photodynamic therapy efficiently regress tumour xenografts in a mouse model of MDA-MB-231 breast cancer. This study suggests the potential of the POLY-PROTACs for precise protein degradation and PROTAC-based cancer therapy.

Heterobifunctional PROteolysis TArgeting Chimeras (PROTACs) hold promising potential for cancer therapy, as they can degrade oncoproteins, particularly undruggable targets[1–3]. PROTACs are generally composed of a warhead that binds to the protein of interest (POI), a ligand-hijacking endogenous E3 ubiquitin ligase, and a linker connecting the warhead and the ligand[4–6]. PROTACs can label the POI with ubiquitin by recognising the E3 ligase and subsequently degrade the POI through the ubiquitin-proteasome system (UPS)[7–9]. Compared to small molecule inhibitors, PROTACs can potentially degrade any intracellular protein, including undruggable targets (e.g., transcription factors and scaffold proteins)[10–12]. Furthermore, PROTACs are potent

agents that can circumvent acquired drug resistance by degrading whole proteins after a short drug exposure time and a low dosage[13,14]. Despite being promising, conventional small molecular PROTACs generally display unfavourable pharmacokinetics and lack tumour specificity, which might cause systemic toxicity due to their nonspecific distribution in normal tissues[15,16]. Thus, it remains a formidable challenge to achieve tumour-specific delivery and potentiate the antitumor potency of conventional PROTACs.

To achieve tumour-targeted delivery of PROTACs, several ligand modification strategies (e.g., antibody-PROTACs, folate-PROTACs and aptamer-PROTAC conjugates) have been

[1]Department of Medical Ultrasound and Center of Minimally Invasive Treatment for Tumor, Shanghai Tenth People's Hospital, Ultrasound Research and Education Institute, School of Medicine, Tongji University, Shanghai 200072, China. [2]State Key Laboratory of Drug Research & Center of Pharmaceutics, Shanghai Institute of Materia Medica, Chinese Academy of Sciences, Shanghai 201203, China. [3]School of Chemistry and Molecular Engineering, East China Normal University, Shanghai 200241, China. [4]School of Chinese Materia Medica, Nanjing University of Chinese Medicine, Nanjing 210023, China. [5]Institute for Advanced Studies (IAS), Wuhan University, Wuhan 430072, China. [6]Department of Ultrasound, Zhongshan Hospital, Institute of Ultrasound in Medicine and Engineering, Fudan University, 200032 Shanghai, China. ✉e-mail: zaxu@chem.ecnu.edu.cn; xu.huixiong@zs-hospital.sh.cn; hjyu@simm.ac.cn

investigated in recent years[17–22]. These decorated PROTACs have shown increased cellular uptake in vitro. In particular, aptamer-PROTAC conjugates have displayed increased tumour accumulation and antitumor potency in vivo compared to conventional PROTACs[22]. Nevertheless, these ligand-PROTAC conjugates suffer from low serum stability, limited tumour penetration and heterogeneous expression of the receptors in different tumour cells and cancer types. Furthermore, opto-PROTACs were developed for ultraviolet light-inducible protein degradation[23,24]. These photoactivatable PROTACs have been demonstrated to spatiotemporally control protein degradation in vitro[25–27]. However, clinical translation of opto-PROTACs is restricted by the poor tissue penetration profile of ultraviolet light with short wavelength (e.g., 360 nm). Therefore, precise PROTAC delivery to the tumour and efficient POI degradation inside tumour cells remain formidable challenges.

In this work we rationally engineer a polymeric PROTAC (POLY-PROTAC) nanoplatform for tumour-targeted degradation of the bromodomain and extraterminal (BET) protein BRD4. We first synthesise four von Hipel-Lindau (VHL)-based small molecular PROTACs and then design a series of reduction-activatable POLY-PROTACs and self-assemble into micellar nanoparticles (NPs) for systemic PROTAC delivery (Fig. 1a). We subsequently engineer a dibenzocyclooctyne (DBCO)-loaded pretargeted NP to enhance intratumoral accumulation and retention of azide-modified POLY-PROTAC NPs via in situ click reaction. Upon internalisation into the tumour cells, the POLY-PROTAC NPs release the PROTAC payload via glutathione (GSH)-mediated reduction of the disulfide bond (Fig. 1b). We demonstrate that the clickable POLY-PROTAC NPs synergistically induce apoptosis of tumour cells when combined with photodynamic therapy (PDT) in a mouse model of MDA-MB-231 breast cancer (Fig. 1c). This study might provide a

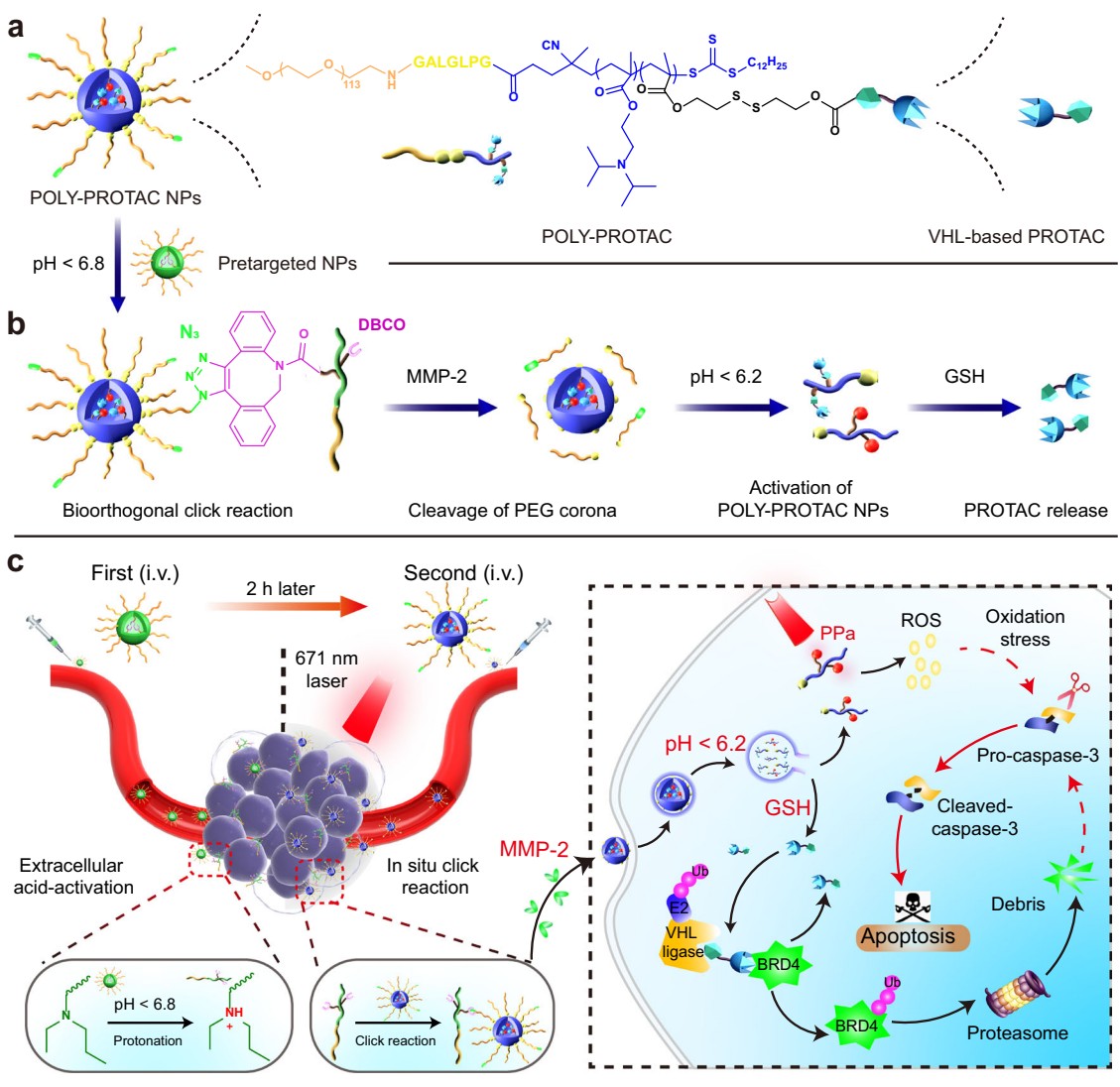

**Fig. 1 | Schematic illustration of the bioorthogonal POLY-PROTAC NPs for tumour-specific protein degradation and precise cancer therapy. a** Cartoon illustration of the azide-functionalized bioorthogonal POLY-PROTAC NPs. POLY-PROTAC was engineered by integrating an MMP-2-liable PEG chain, an acid-activatable DPA moiety and a reduction-sensitive disulfide spacer. **b** Schematic illustration of the extracellular acidity-triggered click reaction between POLY-PROTAC and DBCO-loaded pretargeted NPs and sequential activation of POLY-PROTAC in response to the extracellular enzyme and intracellular acidic/reductive microenvironment. **c** In situ click reaction-promoted protein degradation and combinatorial cancer therapy with POLY-PROTAC NPs. The POLY-PROTAC NPs showed tumour-specific accumulation and retention via a bioorthogonal click reaction with the pretargeted NPs and cleavage of the PEG corona in the tumour mass. The POLY-PROTAC NPs were then internalised into the tumour cells for BRD4 degradation and combination therapy with PDT.

generalisable nanoplatform for tumour-specific PROTAC delivery and potentiated cancer therapy.

## Results

### Synthesis and characterisation of the POLY-PROTAC NPs

BET family proteins, in particular BRD4, have been investigated as promising antitumor targets due to their crucial role in gene transcription[28]. To design a BRD4-targeted POLY-PROTAC, we selected the VHL ligand for PROTAC synthesis since the hydroxyl group of VHL can be reversibly caged via a disulfide spacer. The ability of the modified PROTACs to bind VHL can be restored by GSH-triggered cleavage of the disulfide bond in the cytosol of tumour cells.

We first synthesised four VHL-based PROTACs by adjusting the chemical structures of the VHL ligand and linkers (Fig. 2a,

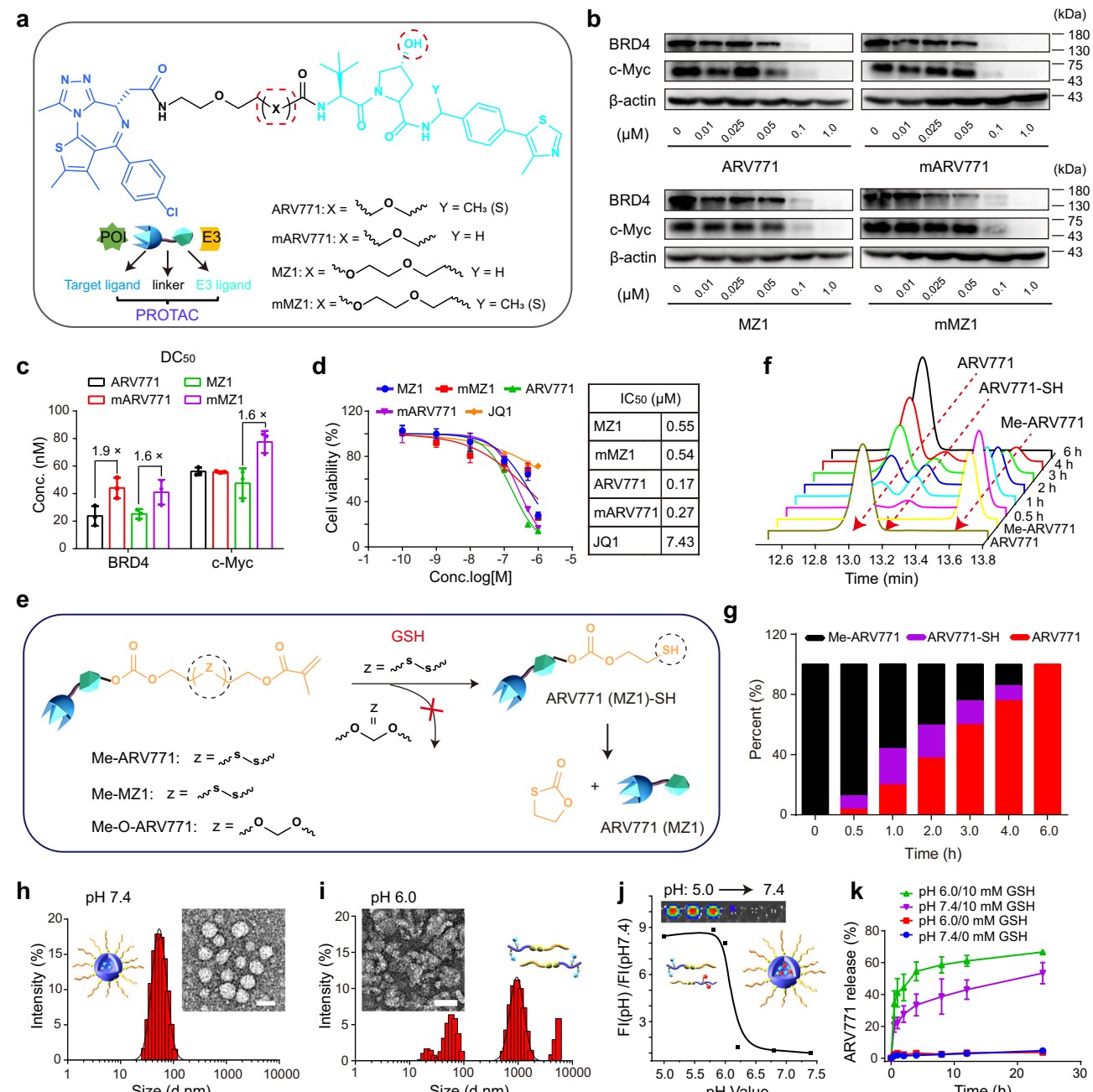

**Fig. 2 | Synthesis and characterisation of the BRD4 PROTAC and POLY-PROTAC NPs. a** Synthesis of the BRD4-targeted VHL PROTACs. **b** Western blot assay of PROTAC-mediated BRD4 degradation and c-Myc downregulation in MDA-MB-231 breast tumour cells after 24 h of incubation. **c** Western blot assay-determined half BRD4 degradation concentration ($DC_{50}$) values of the PROTACs ($n = 3$ biologically independent cells). **d** PROTACs efficiently inhibited the proliferation of MDA-MB-231 cells in a dose-dependent manner in vitro (inset shows the $IC_{50}$ values of the indicated PROTACs and JQ1). Cells were incubated with the PROTACs or JQ1 for 72 h before being subjected to the CCK-8 assay ($n = 4$ biologically independent cells). **e** Schematic illustration of the GSH-triggered activation of ARV771 from Me-ARV771. **f** HPLC chromatograms of the reduction-mediated ARV771 restoration and **g** quantitative ARV771 release percentages from reduction-activatable Me-ARV771 via incubation with 5.0 mM DTT. **h, i** Representative DLS data and TEM images of the PGD7 NPs at **h** pH 7.4 and **i** pH 6.0 (scale bar = 50 μm). **j** Acid-activatable fluorescence profile of PPa-labelled PGDA7 NPs (the fluorescence intensity was normalised to that determined at pH 7.4). Inset shows the fluorescence image of the PGDA7 NP suspensions at different pH values. **k** GSH-triggered ARV771 release from the PGD7 NPs at pH 7.4 and 6.0 (with or without 10 mM GSH addition) ($n = 3$ independent experiments). All data are presented as mean ± SD.

Supplementary Figs. 1–11[29,30]. Western blot assays validated that all four PROTACs remarkably degraded BRD4 in MDA-MB-231 breast cancer cells in vitro and consequently suppressed downstream c-Myc expression. Noticeably, the compounds ARV771 and MZ1 reduced BRD4 expression more efficiently than their analogues. The half-degradation concentration ($DC_{50}$) of ARV771 was 1.9-fold lower than that of mARV771 (Fig. 2b, c and Supplementary Fig. 12). The CCK-8 assay showed that all four PROTACs comparably inhibited the proliferation of MDA-MB-231 cells. Of note, the half inhibition concentrations ($IC_{50}$) of the PROTACs were tens of times lower than that of the BRD4 inhibitor JQ1 (Fig. 2d), suggesting that the BRD4 degraders more significantly impaired the proliferation of the tumour cells than the BRD4 inhibitor.

The hydroxyl group on the VHL ligand plays a crucial role in VHL protein binding by forming hydrogen bonds with HIS-115 and SER-111 in the binding pocket of the VHL protein[31]. The hydroxyl groups of ARV771 and MZ1 were thus methacrylated with a disulfide spacer (termed Me-ARV771 and Me-MZ1) (Fig. 2e) to reversibly disrupt the interaction with the VHL protein and abolish the protein degradation performance of the PROTAC. An ethylene glycol spacer that is inert to reduction was employed to synthesise the GSH-nonresponsive control (namely, Me-O-ARV771). Successful synthesis of the methacrylate-modified PROTACs were validated by proton nuclear magnetic resonance ($^1$H-NMR) and mass spectrometry (MS) measurements (Supplementary Figs. 13–17). High-performance liquid chromatography (HPLC) examination showed that ARV771 was completely regenerated from Me-ARV771 upon 6.0 h of incubation with dithiothreitol (DTT), verifying the superior reduction sensitivity of the disulfide bond-incorporated Me-ARV771 (Fig. 2f, g).

With the reduction-activatable Me-ARV771 and Me-MZ1 in hand, we next sought to synthesise POLY-PROTACs via PEG-induced RAFT copolymerisation of Me-ARV771 (or Me-MZ1) and the acid-ionisable 2-(diisopropylamino)ethyl methacrylate (DPA) monomer for intracellular acidity (e.g., pH 5.5-6.5)[32,33], and reduction-triggered delivery of the BRD4 PROTACs. The resultant POLY-PROTACs were termed according to the PROTAC molecule used: PD7 (based on Me-ARV771) or PDM (based on Me-MZ1). An MMP-2-cleavable GPLGLAG (GG) heptapeptide spacer[34,35], was introduced into the PD7 POLY-PROTAC (namely, PGD7) to promote tumour-specific accumulation and cellular uptake of the POLY-PROTAC NPs (Supplementary Figs. 18–23). Two MZ1-based POLY-PROTACs without the GG peptide (PDM) and with the GG peptide spacer (PGDM) were also synthesised for comparison (Supplementary Figs. 24 and 25). Pheophorbide A (PPa, a well-studied photosensitizer)-modified mPEG$_{113}$-GG-$b$-P(DPA$_{50}$-$r$-HEMA$_5$) (termed PGDA) diblock copolymer was synthesised by grafting PPa onto the pendant hydroxyl groups of PGDH (Supplementary Figs. 26–29). A reduction-insensitive POLY-PROTAC was also synthesised with Me-O-ARV771 (termed PGDO7, Supplementary Fig. 30). All of the synthesised POLY-PROTACs displayed controllable polymerisation and narrow molecular weight distributions as determined by $^1$H-NMR and gel permeability chromatography measurements, respectively (Supplementary Table 1; all NP compositions and acronyms are summarised in Supplementary Fig. 31).

The POLY-PROTAC micellar NPs were then prepared via the nanoprecipitation method as described previously[36,37]. At neutral pH (7.4), dynamic light scattering (DLS) data and transmission electron microscopy (TEM) examinations revealed an average hydrodynamic diameter of ~55 nm and a spherical morphology of the POLY-PROTAC NPs with a narrow particle size distribution (polydispersity index (PDI) < 0.2). In contrast, amorphous aggregates appeared at pH 6.0 due to acid-induced protonation of the DPA groups and dissociation of the POLY-PROTAC NPs (Fig. 2h, i).

To demonstrate the acid-responsive property of the POLY-PROTAC NPs, ARV771- or MZ1-conjugated POLY-PROTAC was then co-assembled with PGDA to obtain fluorophore-labelled POLY-PROTAC NPs (Supplementary Fig. 31). At neutral pH (7.4), the fluorescence of the PPa-labelled POLY-PROTAC NPs was quenched due to homofluorescence resonance energy transfer between PPa molecules[38]. In contrast, the fluorescence emission was remarkably recovered at acidic pH (6.2), which mimics the acidic microenvironment of endosome organelles (pH = 5.8-6.5) (Fig. 2j). This phenomenon further validated the acid-triggered dissociation of the POLY-PROTAC NPs, which might facilitate PROTAC release inside tumour cells.

HPLC examinations showed that in the absence of GSH, ARV771 was marginally released from the PGD7 NPs under both neutral and acidic conditions (pH = 7.4 and 6.0) (Fig. 2k). In contrast, ARV771 was remarkably restored with the addition of 10 mM GSH. For instance, over 50% of the ARV771 was released after 4 h of incubation with 10 mM GSH solution at pH 6.0, which was ~20% higher than that released at neutral pH. This phenomenon could be explained by the increased GSH accessibility when the POLY-PROTAC NPs were dissociated under acidic conditions.

## BRD4 degradation by the POLY-PROTAC NPs in vitro

To demonstrate the advantage of the MMP-2-sheddable POLY-PROTAC for increased cellular uptake and deep tumour penetration (Fig. 3a), PGDA7 (with GG peptide spacer) and PDA7 (without GG peptide spacer) NPs were pretreated with MMP-2 for 1 h to cleave the PEG corona as reported previously[39]. MDA-MB-231 breast tumour cells were then incubated with PDA7 or dePEGylated PGDA7 NPs for the desired time durations. Flow cytometry measurements identified an ~2.8-fold higher intracellular fluorescence signal in the PGDA7 group than in the PDA7 group (Fig. 3b). Confocal laser scanning microscopy (CLSM) examination further revealed remarkably higher intracellular uptake of the PGDA7 NPs than the PDA7 control when examined after 12 h of incubation (Fig. 3c and Supplementary Fig. 32), validating that MMP-2-triggered dePEGylation promoted internalisation of the POLY-PROTAC NPs.

The tumour penetration profile of the PPa-labelled POLY-PROTAC NPs was then investigated in a three-dimensional (3D) multicellular spheroid (MCS) MDA-MB-231 tumour model in vitro. CLSM images showed that after 12 h of incubation, the MMP-2-insensitive PDA7 NPs were primarily entrapped in the peripheral areas and marginally penetrated into the deep areas of the MCSs (Fig. 3d, e). In contrast, the sheddable PGDA7 NPs showed great diffusion into the central areas of the MCSs with ~ 3.0-fold higher fluorescence intensity at a scanning depth of 50 μm (Fig. 3f), implying that the sheddable POLY-PROTAC NPs had a remarkable tumour penetration profile.

With the MMP-2-responsive POLY-PROTAC NPs in hand, we next evaluated their BRD4 degradation performance in vitro. After 24 h of incubation, the GSH-activatable POLY-PROTACs of ARV771 and MZ1 efficiently degraded the BRD4 protein in MDA-MB-231 tumour cells in vitro (Fig. 3g–j). In contrast, the PGDO7 POLY-PROTAC bearing an ethylene glycol linker negligibly affected BRD4 protein expression (Fig. 3k), validating the crucial role of GSH-triggered reduction of the disulfide bond and restoration of the VHL ligand for protein degradation inside the tumour cells. Pretreating the MMP-2-sheddable POLY-PROTAC NPs with MMP-2 remarkably reduced the $DC_{50}$ of the PGD7 and PGDM NPs, which were 1.9- and 3.1-fold lower than their MMP-2-insensitive PD7 and PDM counterparts, respectively, and comparable to those of free ARV771 and MZ1 (Fig. 3l and Supplementary Fig. 33). This could be attributed to the increased cellular uptake of the PGD7 and PGDM NPs due to MMP-2-mediated cleavage of the PEG corona.

Cotreatment with the proteasome inhibitor MG132 abolished the protein degradation ability of the small molecular PROTACs (e.g., MZ1 and ARV771) and the POLY-PROTAC NPs (e.g., PGDM and PGD7) (Fig. 3m), verifying the ubiquitin-proteasome-dependent BRD4 degradation profile of the POLY-PROTAC NPs. The CCK-8 assay further

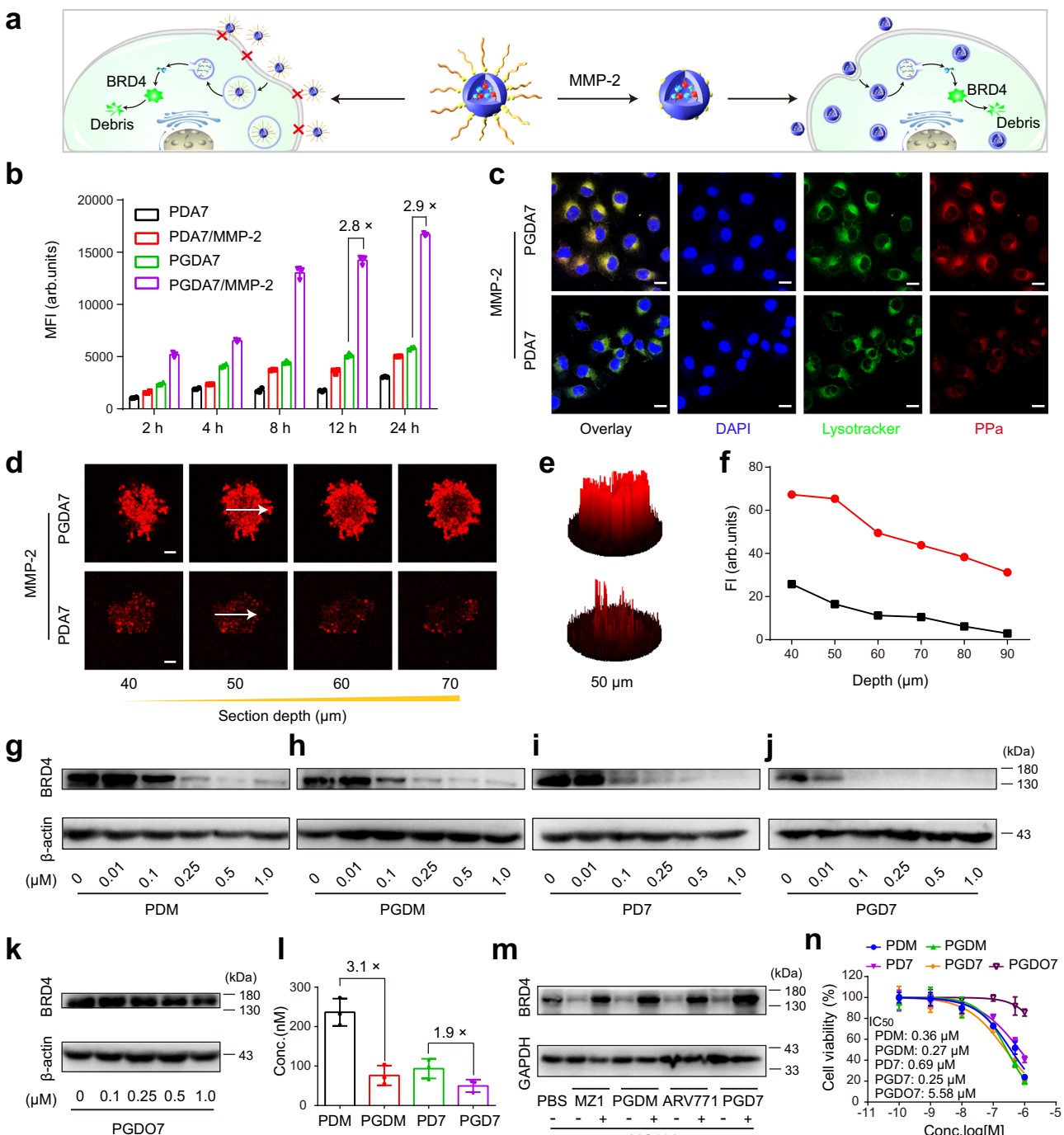

**Fig. 3 | Stimuli-activatable POLY-PROTAC NPs displayed increased cellular uptake, tumour penetration and BRD4 degradation performance in vitro.**
**a** Cartoon illustration of the MMP-2-triggered cleavage of the PEG corona and promoted cellular uptake of the PGD7 NPs in vitro. **b** Flow cytometric analysis of PGDA7 and PDA7 NPs cellular uptake in vitro (the NPs were pretreated with 0.2 mg/mL MMP-2 for 1 h) (n = 3 biologically independent cells). **c** CLSM examination of the intracellular distribution of the POLY-PROTAC NPs after 12 h of incubation (scale bar = 20 μm). **d–f** MMP-2-responsive PGDA7 NPs displayed increased penetration into MDA-MB-231 MCSs in vitro compared with their MMP-2-nonresponsive PDA7 counterpart. **d** CLSM examination of PGDA7 and PDA7 NPs distribution after 12 h of incubation in vitro. **e** 2.5-D reconstruction of the CLSM images at a scanning depth of 50 μm. **f** Fluorescence intensity of the central region versus Z-axis depth. **g–j** Reduction-activatable POLY-PROTAC NPs efficiently degraded the BRD4 protein in MDA-MB-231 cells in vitro. Western blot assay of BRD4 degradation in MDA-

MB-231 cells with various GSH-sensitive POLY-PROTAC NPs in vitro after 24 h of incubation. **k** Western blot assay of BRD4 expression in PGDO7 NP-treated MDA-MB-231 cells in vitro. **l** DC$_{50}$ of the POLY-PROTAC NP-mediated degradation of BRD4 in MDA-MB-231 cells in vitro (n = 4 biologically independent cells). **m** POLY-PROTAC NPs degraded the POI via the ubiquitin-proteasome system. Western blot assay of BRD4 expression in MDA-MB-231 cells with or without MG132 incubation (MZ1/ARV771 concentrations of 1.0 μM and MG132 concentration of 5.0 mM). **n** PGDM and PGD7 NPs efficiently inhibited the proliferation of MDA-MB-231 cells in vitro (n = 4 biologically independent cells) (the POLY-PROTAC NPs were named according to the PROTAC and components integrated. P: PEG chain; G: GPLGLAG peptide; D: acid-activatable DPA group; 7: disulfide bond-bearing ARV771 methacrylate; M: disulfide bond-bearing MZ1 methacrylate; O7: ethylene group-bearing ARV771 methacrylate; A: PPa. See Supplementary Fig. 31 for NPs compositions). All data are presented as mean ± SD.

revealed increased cytotoxicity of the PGDM and PGD7 NPs compared with that of the MMP-2 nonresponsive PGDO7 control after 72 h of incubation (Fig. 3n). Notably, the MMP-2-liable POLY-PROTAC NPs displayed half inhibitory concentrations comparable to that of the parental molecule (Fig. 2d), suggesting that the PROTAC payload can be readily released inside the tumour cells via GSH-mediated cleavage of the disulfide bond. Collectively, the above data demonstrated that the POLY-PROTAC NPs with sheddable PEG coronas and reduction-liable linkers efficiently degraded the POI and suppressed tumour cell proliferation in vitro.

## Biodistribution and antitumor performance of the POLY-PROTAC NPs in vivo

The tumour-specific accumulation and penetration of the POLY-PROTAC NPs were next investigated in an MDA-MB-231 breast tumour-bearing BALB/c nude mouse model in vivo. MMP-2-sheddable PGDA7 and the MMP-2-insensitive PDA7 analogue NPs were administered via intravenous (i.v.) injection at an identical ARV771 dose of 10 mg/kg and PPa dose of 5.0 mg/kg when the tumour volume reached 200 mm³. In vivo fluorescence imaging displayed clear tumour distribution of the PGDA7 NPs over time (Fig. 4a), which can be explained by the

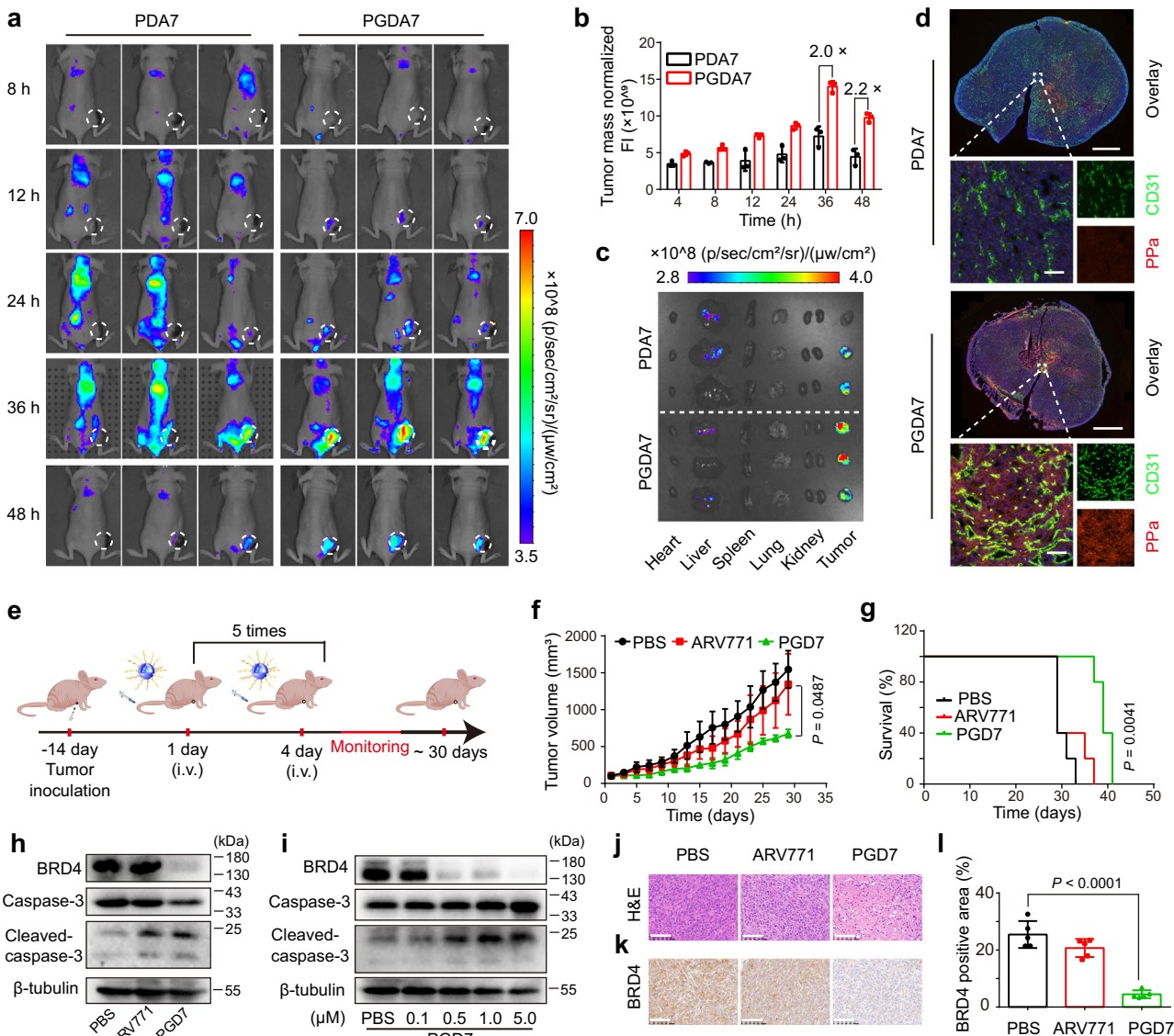

**Fig. 4 | Stimuli-activatable POLY-PROTAC NPs specifically accumulated at the tumour site and suppressed tumour growth in vivo. a** Fluorescence images of POLY-PROTAC NPs distribution in MDA-MB-231 tumour-bearing nude mice in vivo. MMP-2-activatable PGDA7 NPs specifically accumulated at the tumour site. **b** Normalised fluorescence intensities of the tumour tissue ($n = 3$ biologically independent mice). **c** Ex vivo fluorescence images of the major organs and tumour (the major organs and tumours were harvested 48 h post-injection). **d** Ex vivo CLSM images of tumour sections at 48 h post-injection (top panel scale bar = 2.0 mm, bottom panel scale bar = 50 μm). **e** Treatment schedule of the POLY-PROTAC NPs antitumor study in MDA-MB-231 tumour-bearing nude mice. **f, g** PGD7 NPs efficiently suppressed MDA-MB-231 tumour growth without notable adverse effects. **f** Tumour growth curves and **g** survival plots of the tumour-bearing BALB/c nude mice upon POLY-PROTAC NPs treatment ($n = 5$ biologically

independent mice) Statistical analysis of **f** was performed by two-sided unpaired t-test. Statistical significance of **g** was calculated by survival curve comparison with Log-rank (Mantel-Cox) test. **h, i** Western blot assay of PGD7 NP-induced BRD4 degradation and caspase-3 activation in **h** MDA-MB-231 tumours in vivo (the tumours were harvested on the second day after five cycles of treatment), and **i** tumour cells in vitro as a function of ARV771 concentration. **j** H&E staining of the tumour sections at the end of the antitumor study (scale bar = 100 μm). **k–l** PGD7 NPs degraded the BRD4 protein and activated caspase-3 in vivo. **k** Immunohistochemical staining of the BRD4 protein (scale bar = 100 μm) and **l** Semiquantitation of IHC-determined BRD4 expression in the tumour sections ($n = 5$ biologically independent mice). Statistical analysis was performed by two-sided unpaired t-test. All data are presented as mean ± SD.

enhanced permeability and retention effect of the NPs[40]. Remarkably, the PGDA7 NPs showed much higher intratumoural accumulation and slower blood clearance than the MMP-2-insensitive PDA7 control at all examined time points. For instance, the tumour fluorescence in the PGDA7 group was 2.0-fold higher than that in the PDA7 control group at 36 h post-injection (Fig. 4b).

The tumour-specific distribution of the PGDA7 NPs was confirmed by ex vivo fluorescence imaging of the major organs (e.g., heart, liver, spleen, lung and kidney) and the tumour tissue at 48 h post-injection (Fig. 4c). CLSM examination of the tumour section further illustrated that the PDA7 NPs were distributed in the peripheral area of the blood vessels. In contrast, the PGDA7 NPs diffused throughout the tumour tissue (Fig. 4d). Taken together, the fluorescence imaging and CLSM examination data verified increased tumour accumulation and penetration of the PGDA7 NPs via MMP-2-mediated cleavage of the PEG corona.

We subsequently explored the antitumor efficacy of the POLY-PROTAC NPs in an MDA-MB-231 tumour model in vivo. The tumour-bearing BALB/c nude mice were randomly grouped when the tumour volumes reached 100 mm$^3$ and i.v. injected with PBS, ARV771 or the PGD7 NPs at an identical ARV771 dosage of 10 mg/kg. The treatments were repeated every three days for a total of five injections (Fig. 4e). Free ARV771 negligibly affected MDA-MB-231 tumour growth compared to the PBS control. In contrast, the PGD7 NPs significantly delayed tumour growth by ~50% and consequently prolonged the survival of the tumour-bearing mice (Fig. 4f, g).

To elucidate the mechanism underlying the antitumor performance of the POLY-PROTAC NPs, BRD4 degradation in lysates from the tumours of the PGD7 group were examined by western blot assay. Free ARV771 displayed a negligible influence on BRD4 expression in tumour xenografts. In contrast, PGD7 NPs remarkably suppressed BRD4 expression in vivo by ~80% in the tumour tissue (Fig. 4h and Supplementary Fig. 34a) due to the improved delivery efficacy of ARV771 from the POLY-PROTAC NPs. Notably, PGD7 NPs cleaved caspase-3 1.5-fold more efficiently than ARV771 in the tumour tissue in vitro (Fig. 4h and Supplementary Fig. 34b), implying apoptosis of the tumour cells since caspase-3 is a crucial executor of cellular apoptosis[41]. Noticeably, free ARV771 and the PGD7 NPs comparably activated caspase-3 in MDA-MB-231 tumour cells (Fig. 4i and Supplementary Fig. 34c–e). Therefore, the PGD7 NP-mediated caspase-3 activation profile could be attributed to the BRD4 degradation property of ARV771.

BRD4 degradation-induced apoptosis of tumour cells in vivo was verified by haematoxylin-eosin (H&E) staining of the tumour sections (Fig. 4j). Immunohistochemistry (IHC) examination of the tumour sections showed that free ARV771 negligibly affected BRD4 expression (Fig. 4k). In contrast, PGD7 NP remarkably suppressed BRD4 expression with an ~ 4-fold lower BRD4-positive area than free ARV771 (Fig. 4l). The experimental groups displayed similar body weight changes during the whole experimental period compared to the PBS group (Supplementary Fig. 35a). Furthermore, the H&E images revealed negligible histopathological damage to the major organs (e.g., heart, liver, spleen, lung and kidney) (Supplementary Fig. 35b), verifying that the PGD7 POLY-PROTAC NPs had good biosafety.

## The bioorthogonal click reaction amplified the tumour distribution of the POLY-PROTAC NPs in vivo

Insufficient tumour distribution of PROTACs is one of the bottlenecks for PROTAC-based cancer therapy. To further prompt tumour-specific accumulation of the POLY-PROTAC NPs via a copper-free click reaction in the biological milieu[42], we next designed an extracellular tumour acidity-activatable pretargeted NPs for tumour-targeted delivery of the dibenzocyclooctyne (DBCO) groups (Fig. 5a). The pretargeted NPs were prepared by the self-assembly of DBCO-modified mPEG-b-poly (ethylene propyl amine) (namely, PED) diblock copolymer (Supplementary Figs. 36 and 37 for PED diblock copolymer synthesis).

The PED pretargeted NPs displayed a homogeneous and spherical morphology with an average hydrodynamic diameter of ~60 nm at neutral pH (i.e., 7.4) (Fig. 5b) and with dramatic dissociation at the acidic pH of 6.5 (Fig. 5c), which mimics the acidic tumour microenvironment (e.g., pH = 6.5–6.8)[33]. The fluorescence of the PPa-conjugated PED NPs (Supplementary Fig. 38 for polymer synthesis) was quenched under neutral conditions. In contrast, the fluorescence of PPa was recovered at pH values below 6.6 via the extracellular acid-induced dissociation of the pretargeted NPs (Fig. 5d), validating the superior extracellular acid-responsive property of the PED NPs.

Subsequently, an azide group was modified on the hydrophilic PEG head of the PGDH diblock copolymer (N$_3$PGDH) (Supplementary Fig. 39) to prepare azide-functionalized PGD7 POLY-PROTAC (namely, N$_3$@PGD7) NPs. The azide group on the surface of the N$_3$@PGD7 NPs was expected to perform in-situ click reaction with DBCO groups in the acidic tumour microenvironment. Both the PED and N$_3$@PGD7 NPs displayed good serum stability in 10% (V/V) foetal bovine serum-containing neutral buffer solution (pH 7.4) (Supplementary Fig. 40). DLS examination showed a uniform hydrodynamic diameter and narrow particle size distribution when the PED NPs and the N$_3$@PGD7 POLY-PROTAC NPs were mixed under neutral pH conditions (Fig. 5e), implying that the PED and N$_3$@PGD7 NPs could remain inert during blood circulation to avoid unfavourable interactions by encapsulating the DBCO groups inside the hydrophobic core of the PED NPs. In contrast, DLS and TEM examinations revealed amorphous aggregates with a broad size distribution when the PED and N$_3$@PGD7 NPs were coincubated at an acidic pH of 6.5 (Fig. 5f), validating the occurrence of a multivalent click reaction between the DBCO-grafted PED copolymer and azide-modified POLY-PROTAC NPs to form cross-linked nanostructures.

In vivo fluorescence imaging displayed a clear intratumoural fluorescence signal when the PED pretargeted NPs were i.v. injected into MDA-MB-231 tumour-bearing nude mice (Fig. 5g), which was caused by tumour acidity-triggered protonation of the tertiary amine of EPA and subsequent dissociation of the PED NPs. CLSM examination of the tumour sections showed that the PED pretargeted NPs colocalized well with the cell membrane (labelled with wheat germ agglutinin (WGA)) 2-4 h post-injection, verifying that the PED NPs dissociated and exposed the DBCO groups in the extracellular matrix (ECM) of the tumour tissue (Fig. 5h); this facilitates the click reaction between the DBCO and the azide groups on the surface of the POLY-PROTAC NPs in the ECM.

To evaluate whether the in situ click reaction enhances tumour-targeted PROTAC delivery in vivo, N$_3$@PGDA7 NPs integrating ARV771 POLY-PROTAC and PPa were prepared by coassembling N$_3$@PGDH, PGD7 and PGDA diblock copolymers (Supplementary Fig. 31), which was i.v. administered 2 h after pretargeted NP injection. The PED + N$_3$@PGDA7 group displayed a much brighter intratumoural fluorescence signal than that observe in the N$_3$@PGDA7-injected mice (Fig. 5i). For instance, the PED + N$_3$@PGDA7 group showed an ~ 2.0-fold higher fluorescence signal than the N$_3$@PGDA7 NP group at 24-36 h post-injection (Fig. 5k).

The increased tumour distribution of the POLY-PROTAC NPs was validated by the ex vivo fluorescence imaging of the major organs and tumour tissues 48 h post-injection (Fig. 5l). CLSM examination of the tumour sections demonstrated that the pretargeted PED NPs dramatically increased tumour accumulation and penetration of the N$_3$@PGDA7 POLY-PROTAC NPs (Fig. 5j). HPLC examination of ARV771 distribution in vivo further demonstrated increased intratumoural distribution and retention of the POLY-PROTAC NPs via an in situ click

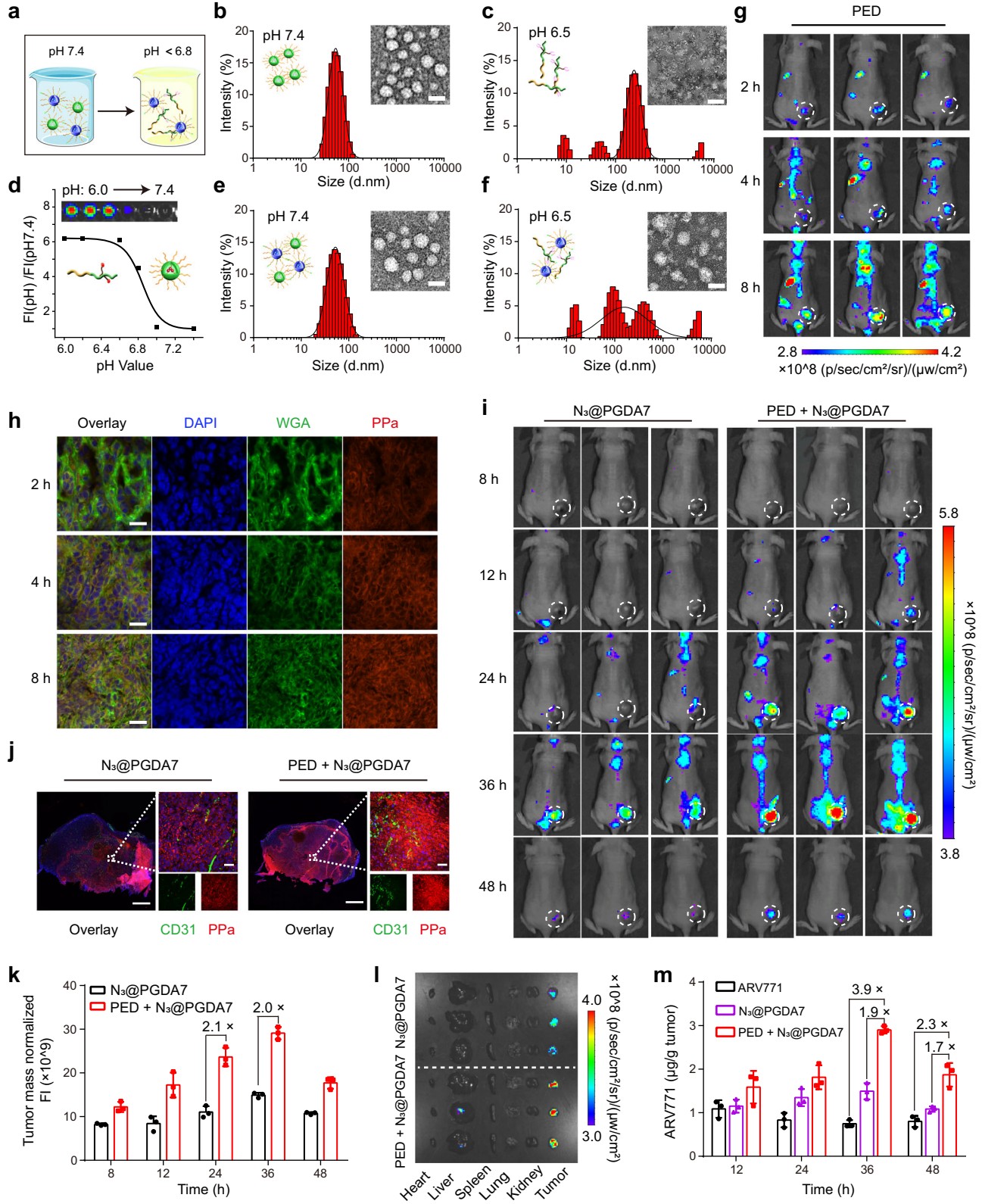

reaction. In comparison with free ARV771 and N₃@PGDA7 NPs, the combination of PED + N₃@PGDA7 NP remarkably enhanced the tumour distribution of ARV771 by 3.9- and 1.9-fold, respectively, when examined at 36 h post-injection (Fig. 5m). The data from both the fluorescence imaging and HPLC analyses provided consistent evidence that the in situ click reaction significantly enhanced the accumulation and retention of ARV771 PROTAC at the tumour site.

## Bioorthogonal POLY-PROTAC NPs regressed breast tumour growth in vivo

Given the significantly improved tumour distribution of the clickable POLY-PROTAC NPs, we next investigated their antitumor performance in the in vivo MDA-MB-231 tumour model (Fig. 6a). Compared to ARV771 or N₃@PGD7 NPs alone, PED + N₃@PGD7 much more efficiently delayed ~70% of the MDA-MB-231 tumour growth due to

**Fig. 5 | An in situ click reaction amplified the tumour accumulation of the bioorthogonal POLY-PROTAC NPs in vivo. a** Diagrammatic illustration of the tumour-specific delivery of the POLY-PROTAC NPs via a bioorthogonal click reaction with the extracellular tumour acidity-activatable pretargeted NPs. **b–d** DLS and TEM examinations of the pretargeted NPs at **b** pH 7.4 and **c** pH 6.5 (scale bar = 50 μm). The PED pretargeted NPs dissociated in the acidic extracellular tumour environment and exposed DBCO in vitro. **d** Normalised fluorescence plots of the pretargeted NPs versus pH value (inset: fluorescence image of the PEA NPs under different pH conditions). **e, f** DLS and TEM examinations of the particle distribution and morphology of the mixture of PED + $N_3$@PGD7 NPs at **e** pH 7.4 and **f** 6.5 (scale bar = 50 μm). The bioorthogonal click reaction occurred between the PED pretargeted NPs and the $N_3$@PGD7 POLY-PROTAC NPs at pH 6.5. **g, h** The pretargeted NPs specifically accumulated at the tumour site and were activated by the acidic tumour pH. **g** Fluorescence images of the biodistribution of PED NPs in the MDA-MB-231 tumour-bearing nude mice in vivo and **h** ex vivo CLSM examination of tumour sections (scale bar = 50 μm). The experiment was repeated independently 3 times with similar results. **i–m** PED NPs increased the tumour distribution of $N_3$@PGD7 NPs after the bioorthogonal click reaction in vivo. **i** Fluorescence imaging of MDA-MB-231 tumour-bearing BALB/c nude mice and **j** CLSM images of the tumour sections at 48 h post-injection (scale bar = 50 μm) (the mice were i.v. injected with the PED pretargeted NPs at a DBCO dose of 1.0 mg/kg and subsequently i.v. injected with the $N_3$@PGDA7 NPs at an azide dose of 0.055 mg/kg at 2 h after PED injection). The in situ bioorthogonal click reaction markedly increased ARV771 distribution in the tumour. **k** Intratumoural fluorescence intensity in $N_3$@PGDA7 NP-injected mice (*n* = 3 biologically independent mice). **l** Fluorescence images of the major organs 48 h post-injection. **m** HPLC-determined intratumoural ARV771 distribution (*n* = 3 biologically independent mice). All data are presented as mean ± SD.

increased ARV771 distribution in the tumour tissue (Fig. 6b). However, PED + $N_3$@PGD7 marginally prolonged the survival time of the tumour-bearing mice compared to treatment with the $N_3$@PGD7 NPs due to tumour relapse during the late stage of the antitumor study (Fig. 6c).

In previous studies, we explored intracellular acid-activatable PDT to circumvent multidrug resistance of breast tumours[43]. Herein, we next sought to explore the potential of the bioorthogonal POLY-PROTAC NPs for PDT-promoted BRD4 degradation therapy (Fig. 6d). Upon 671 nm laser irradiation, the ARV771-free PGDA NPs remarkably induced the generation of reactive oxygen species and activated the caspase-3 protein in MBA-MB-231 tumour cells in vitro, validating the photoactivity of the PPa-labelled NPs (Supplementary Figs. 42 and 43). The combination of Laser + PGDA7 NPs activated the caspase-3 protein in MDA-MB-231 tumour cells more efficiently than PDT (Laser + PGDA) or BRD4 degradation with PGDA7 alone (Fig. 6e), implying the improved apoptosis induction performance of PDT combined with BRD4 degradation in vitro.

The antitumor performance of combining PDT with BRD4 degradation was next investigated in vivo. MDA-MB-231 tumour-bearing BALB/c nude mice were randomly grouped when the tumour volumes reached ~ 100 mm³, and they were then treated with PBS, ARV771, PED + $N_3$@PGDA + Laser, PED + $N_3$@PGD7, PED + PGDA7 + Laser, or PED + $N_3$@PGDA7 + Laser (Fig. 6f). The $N_3$@PGDA7 NPs were i.v. administered 2 h post-injection of the PED NPs, and 671 nm laser irradiation was applied 36 h post-injection of the $N_3$@PGDA7 NPs. Figure 6g demonstrates that PDT or ARV771 administration alone marginally suppressed the proliferation of MDA-MB-231 tumours. In contrast, the combination of the bioorthogonal NPs (PED + $N_3$@PGDA7) and PDT dramatically regressed 95% of tumour growth, which was 1.5-fold more efficient than BRD4 degradation alone caused by PED + $N_3$@PGDA7 treatment.

Furthermore, the combination of PED pretargeted NPs with $N_3$@PGDA7 and PDT prolonged the survival of the tumour-bearing mice by 40% compared to those in the PED + $N_3$@PGDA7 group, with 40% of the animals surviving 100 days post-treatment (Fig. 6h). TUNEL staining of the tumour sections revealed notable apoptosis of the tumour cells from the mice in the PED + $N_3$@PGDA7 + PDT group, suggesting that the combination of PDT and BRD4 degradation with the bioorthogonal NPs cumulatively induced apoptosis of the tumour cells in vivo (Fig. 6j). Semiquantitative analysis of the TUNEL staining data further revealed that PED + $N_3$@PGDA7 + Laser treatment more efficiently induced apoptosis of the tumour cells than ARV771 and PED + $N_3$@PGDA7 by 15.2- and 4.2-fold, respectively (Supplementary Fig. 44). Treatment-induced apoptosis of the tumour cells was further confirmed by ex vivo H&E staining of the tumour sections (Fig. 6k). Moreover, negligible body weight loss and histopathological damage to the major organs were observed during the experimental period, verifying the satisfactory biosafety of the POLY-PROTAC NPs (Fig. 6i and Supplementary Fig. 44).

Immunohistochemical examination of the tumour tissue demonstrated significantly decreased BRD4 expression in vivo (Fig. 6l). Semiquantitative analysis of the IHC images further determined that PED + $N_3$@PGDA7 treatment more efficiently downregulated BRD4 expression (by 2.6-fold) than $N_3$@PGDA7 NP alone, validating that the increased intratumoural distribution of ARV771 via the in situ bioorthogonal reaction contributed to BRD4 degradation in vivo (Fig. 6m). Notably, the combination of PDT and PED + $N_3$@PGDA7 further promoted BRD4 degradation compared to PED + $N_3$@PGDA7 treatment due to PDT-enhanced intracellular release of the ARV771 PROTAC in vivo (Fig. 6m, n). Additionally, combinatorial therapy with PED + $N_3$@PGDA7 + Laser more efficiently activated caspase-3 (1.8-fold) and degraded BRD4 (3.0-fold) than PED + $N_3$@PGD7, verifying the synergistic apoptosis-induction profile caused by combined PDT and BRD4 degradation (Fig. 6o).

## Discussion

Heterobifunctional PROTACs with the ability to degrade proteins have been recently exploited for cancer therapy[4]. However, the clinical translation of small molecular PROTACs suffers from low bioavailability and insufficient tumour specificity. It is therefore highly desirable to develop novel PROTACs for tumour-specific protein degradation while minimising on-target, off-tumour adverse effects. A tumour microenvironment-activatable POLY-PROTAC nanoplatform was thus engineered herein for tumour-specific delivery and potentiation of the antitumor performance of PROTACs.

In comparison with their small molecular PROTAC counterparts, the POLY-PROTAC NPs reported in this study possess several distinct advantages. First, the POLY-PROTAC NPs with MMP-2-liable PEG coronas elongated the blood circulation of the small molecular PROTACs, while the PEG coronas were cleaved via intratumoural MMP-2 to facilitate PROTAC tumour-specific accumulation and retention. Second, the POLY-PROTAC NPs can disintegrate in the intratumoural acidic microenvironment. The PROTAC prodrug can thus be restored in the cytosol via GSH-mediated cleavage of the disulfide bond. Furthermore, we demonstrated that the POLY-PROTAC NPs can be adapted for bioorthogonal click reaction-enforced tumour specificity. Remarkably, the azide-modified POLY-PROTAC NPs showed 3.9-fold higher tumour accumulation than their small molecule counterparts via an in situ click reaction with the pretargeted NPs, which therefore further boosted PROTAC-based cancer therapy. Moreover, other kinds of stimuli-activatable chemical bonds (e.g., thioketone, selenic or boric acid bonds)[44,45] can be utilised to achieve spatial-temporal PROTAC activation. Apart from PDT, the POLY-PROTAC nanoplatform can be combined with other kinds of therapeutic modalities (e.g., radiotherapy or chemotherapy), and these are both under investigation in our laboratory for potentiated antitumor performance.

In summary, a POLY-PROTAC prodrug strategy was developed for the tumour-specific delivery of PROTACs in this study. We

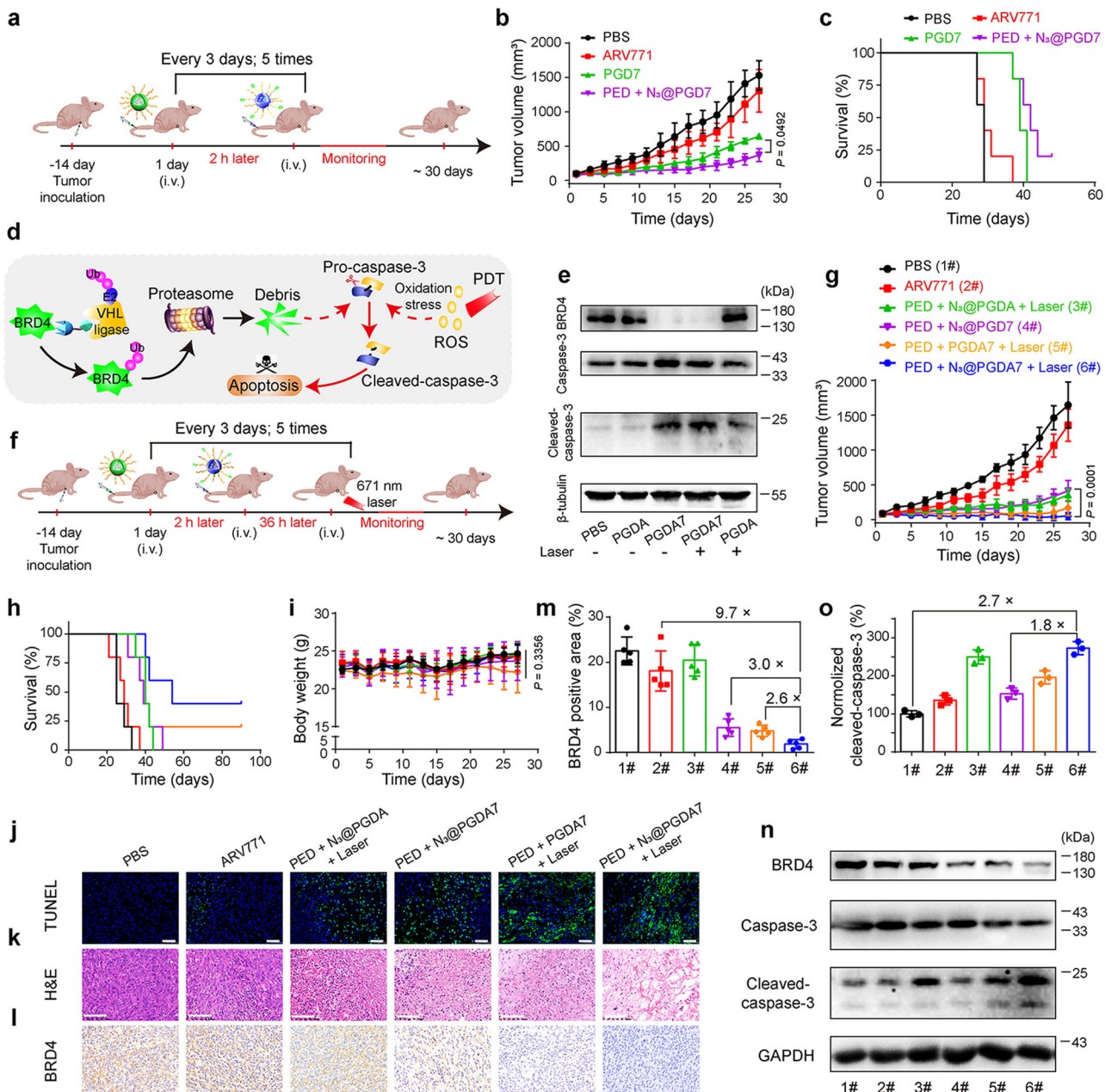

**Fig. 6 | Combining the bioorthogonal POLY-PROTAC NPs with PDT efficiently regressed breast tumour growth in vivo. a** Experimental schedule for the bioorthogonal POLY-PROTAC NP-based antitumor study. **b** Averaged tumour growth curves after treatment with various formulations (PBS, ARV771, PGD7 and PED + N₃@PGD7; $n = 5$ biologically independent mice). Statistical analysis was performed by two-sided unpaired $t$-test. **c** Survival curves of the tumour-bearing mice ($n = 5$ biologically independent mice). **d** Schematic illustration of bioorthogonal POLY-PROTAC NP-based combinatorial therapy of breast tumours via BRD4 degradation and PDT-induced caspase-3 activation in the tumour cells. **e** The combination of ARV771 and PDT cumulatively activated caspase-3 in the tumour cells. Western blot assay of the tumour lysates after different treatments. **f** Experimental schedule of bioorthogonal POLY-PROTAC NP-performed combinatory therapy in an MDA-MB-231 xenograft tumour model. **g** Averaged tumour growth curves after treatment with various formulations (PBS, ARV771, PED +

N₃@PGDA + Laser, PED + N₃@PGD7, PED + PGD7 + Laser, PED + N₃@PGDA7 + Laser; $n = 5$ biologically independent mice; irradiation conditions: 400 mW/cm², 5.0 min). Statistical analysis was performed by two-sided unpaired $t$-test. **h** Survival curves and **i** body weight changes of the tumour-bearing mice during the experimental period ($n = 5$ biologically independent mice). Statistical analysis was performed by one-way ANOVA with a Brown-Forsythe test. **j** TUNEL (blue: DAPI, green: apoptotic cells) and **k** H&E staining of the tumour sections 30-days post-treatment (scale bar = 100 μm). **l** IHC analysis and **m** Semiquantitative BRD4 expression in the tumour tissues ($n = 5$ biologically independent mice). **n** Western blot assay and **o** normalised expression of activated caspase-3 in the tumour lysates ($n = 3$ biologically independent mice) (1#: PBS; 2#: ARV771; 3#: PED + N₃@PGDA + Laser; 4#: PED + N₃@PGD7; 5#: PED + PGD7 + Laser; 6#: PED + N₃@PGDA7 + Laser). All data are presented as mean ± SD.

demonstrated that this POLY-PROTAC nanoplatform can be activated by multiple stimuli (extracellular MMP-2, intracellular acidity and reductive conditions), which can achieve increased tumour accumulation, deep tumour penetration and enhanced protein degradation performance compared with its small molecule counterpart. The

POLY-PROTAC nanoplatform with an extracellular tumour acidity-triggered bioorthogonal click reaction remarkably enhanced tumour-specific PROTAC delivery. This clickable POLY-PROTAC nanoplatform can be further applied with PDT for protein degradation and combination therapy, as this combinatorial treatment suppressed over 95%

of MDA-MB-231 TNBC tumour growth. The extracellular acidic microenvironment containing MMP-2 has been well documented in various solid tumours. Therefore, this acid-activatable bioorthogonal POLY-PROTAC nanoplatform can be further employed for combination therapy to treat a broad spectrum of cancers by integrating multiple therapeutic regimens (e.g., chemotherapeutics and immunotherapeutics). Taken together, the proposed POLY-PROTAC nanoplatform might provide a novel avenue to potentiate PROTAC-based cancer therapy.

# Methods
## Materials
1-(3-Dimethylaminopropyl)−3-ethylcarbodiimide hydrochloride (EDCI), 1-Hydroxybenzotriazole (HOBT), 4-Dimethylaminopyridine (DMAP), Trifluoroacetic acid (TFA), N, N-Diisopropylethylamine (DIEA), N,N-Dimethylacetamide (anhydrous, DMAC), Dichloromethane (anhydrous, DCM), Dimethyl sulfoxide (anhydrous, DMSO), N,N-Dimethylformamide (anhydrous, DMF), 2,2′-Azobis (2-methylpropionitrile (AIBN), and methanol (HPLC, MeOH) were purchased from J&K Chemicals (Shanghai, China). 4-Cyano-4-(dodecylsulfanylthiocarbonyl) sulfanylpentanoic acid (CDP), 2-(diisopropylamino) ethyl methacrylate (DPA), IV Gelatinase, Triethylene glycol and tert-butyl (S)−2-(4-(4-chlorophenyl)−2,3,9-trimethyl-6H-thieno[3,2-f][1,2,4]triazolo[4,3-a][1,4]diazepin-6-yl)acetate (JQ1) were obtained from Sigma-Aldrich (Shanghai, China). Bis(2-hydroxyethyl) disulfide, Triphosgene, 2-bromoethanol, 2-hydroxyethyl methacrylate, Methacryloyl chloride and methacryloyl chloride N-ethylpropan-1-amine were purchased from TCI (Shanghai, China). Tert-butyl 2-(3-(2-aminoethoxy)propoxy) acetate, tert-butyl-2-(2-(2-(2-aminoethoxy)ethoxy)ethoxy) acetate were all purchased from Shanghai Tebo Chemical Technology Co., LTD. Fmoc-protected heptapeptide Gly-Pro-Leu-Gly-Leu-Ala-Gly (Fmoc-GPLGLAG) was synthesised by GL Biochem. Co., Ltd (Shanghai, China). Methoxy poly(ethylene glycol) amine (mPEG$_{113}$-NH$_2$) and azide-poly(ethylene glycol) amine (N$_3$-PEG$_{113}$-NH$_2$) were purchased from Seebio Biotech. Co., Ltd (Shanghai, China). Pyropheophorbide a (PPa) and Bis(4-nitrophenyl) carbonate were purchased from Dibai biotechnology CO., LTD. (Shanghai, China). Tert-butyl ((2 S)−1-((4 R)−4-hydroxy-2-(((S)−1-(4-(4-methylthiazol-5-yl)phenyl)ethyl)carbamoyl)pyrrolidin-1-yl)−3,3-dimethyl-1oxobutan −2-yl) carbamate was obtained from Ruozhi Chemical Technology Co., LTD. (Shanghai, China).

Dulbecco's modified eagle medium (DMEM), 0.25% trypsin-EDTA, penicillin/streptomycin solution, PBS buffer solution (1×), bovine serum albumin (BSA, fraction V), BCA protein quantification kit, protein marker (10-190 kDa), TBST buffer solution (10×), HEPES buffer solution (2×), 4′,6-Diamidino-2-phenylindole dihydrochloride (DAPI), Anti-fade solution, Lyso-green DND-189, singlet oxygen fluorescent probe SOSG, reactive oxygen species fluorescent probe DCFH-DA, cell counting kit-8 (CCK-8), MG132 proteasome inhibitor and ECL luminescence reagent were purchased from Meilun Biotech Co., Ltd (Dalian, China). Foetal bovine serum (FBS) was purchased from Gibco (Tulsa, OK). BD Matrigel, Tris-Base, glycine, SDS, Peroxidase-Conjugated Goat Anti-Rabbit IgG (H + L) (CAT:33101ES60) and Peroxidase-Conjugated Goat Anti-Mouse IgG (H + L) (CAT:33201ES60) were purchased from YEASEN (Shanghai, China). The anti-BRD4 antibody (ab128874), anti-c-Myc antibody (ab32072), anti-caspase-3 antibody (ab 184787), anti-GAPDH antibody (ab8245), anti-β-actin antibody (ab8226), anti-β-tubulin antibody (ab78078) were all purchased from Abcam (Shanghai). All other reagents and solvents were analytical grade and obtained from SinoPharm Chemical Reagent Co., Ltd. (Shanghai, China).

## Cell lines and animals
MDA-MB-231 human breast cancer cell line was obtained from the cell bank of Chinese Academy of Sciences (Shanghai, China). The cells were incubated in complete DMEM cell culture medium containing FBS 10% (V/V), penicillin G sodium and streptomycin sulfate (100 U/mL), and incubated at 37 °C under a humidified atmosphere containing 5% of CO$_2$. All cell culture studies were performed in the logarithmic phase of cell growth in vitro.

Balb/c nude mice (female, 4-week old, 18 - 20 g) were obtained from Shanghai Experimental Animal Center (Shanghai, China). Animals were housed under SPF conditions in groups of 4-5 mice per cage, and maintained at a temperature of ~25 °C in a humidity-controlled environment with a 12 h light/dark cycle, with free access to standard food and water. All animal procedures were carried out under the guidelines approved by the Institutional Animal Care and Use Committee (IACUC) of Shanghai Institute of Material Medica, Chinese Academy of Sciences.

## Preparation of the POLY-PROTAC and pretargeted nanoparticles
Briefly, mPEG$_{113}$-GALGLPG-$b$-P(DPA$_{42}$-$r$-ARV771$_2$) (2.0 mg) and mPEG$_{113}$-GALGLPG-$b$-P(DPA$_{48}$-$r$-HEMA$_7$) (1.0 mg) were dissolved in 100 μL of DMF, and the mixture was added into 1 mL of DI water under ultrasonication to obtain the POLY-PROTAC nanoparticles (termed as PGD7 nanoparticles). The organic solvent was removed by dialyzing against DI water. One set of the POLY-PROTAC nanoparticles were prepared by following the same procedure described above:

POLY-ARV77 nanoparticles (PD7): mPEG$_{113}$-$b$-P(DPA$_{46}$-$r$-ARV771$_2$) + mPEG$_{113}$-$b$-P(DPA$_{52}$-$r$-HEMA$_6$);

MMP-2-liable POLY-ARV77 nanoparticles (PGDA7): mPEG$_{113}$-GALGLPG-$b$-P(DPA$_{46}$-$r$ ARV771$_2$) + mPEG$_{113}$-GALGLPG-$b$-P(DPA$_{48}$-$r$-PPa$_4$);

PPa-modified POLY-ARV771 nanoparticles (PDA7): mPEG$_{113}$-$b$-P(DPA$_{46}$-$r$-ARV771$_2$) + mPEG$_{113}$-$b$-P(DPA$_{52}$-$r$-PPa$_4$);

POLY-MZ1 nanoparticles (PDM): mPEG$_{113}$-$b$-P(DPA$_{42}$-$r$-MZ1$_2$) + mPEG$_{113}$-$b$-P(DPA$_{52}$-HEMA$_6$);

MMP-2-liable POLY-MZ1 nanoparticles (PGDM): mPEG$_{113}$-GALGLPG-$b$-P(DPA$_{46}$-$r$-MZ1$_2$) + mPEG$_{113}$-GALGLPG-$b$-P(DPA$_{48}$-$r$-HEMA$_7$);

DBCO-loaded pretargeted nanoparticles (PEA): mPEG$_{113}$-$b$-P(EPA$_{60}$-$r$-DBCO$_4$) (namely PED) and mPEG$_{113}$-$b$-P(EPA$_{60}$-$r$-PPa$_1$);

To prepare the azide-functionalized POLY-ARV771 nanoparticles (N$_3$@PGDA7): mPEG$_{113}$-GALGLPG-$b$-P(DPA$_{42}$-$r$-ARV771$_2$) (1.7 mg) + mPEG$_{113}$-GALGLPG-$b$-P(DPA$_{48}$-$r$-PPa$_4$) (0.85 mg) + N$_3$-PEG$_{113}$-GALGLPG-$b$-P(DPA$_{41}$-$r$-mPEG$_{113}$-GALGLPG-$b$-P(DPA$_{48}$-$r$-PPa$_4$) (0.45 mg) were dissolved into 100 μL of DMF, and added into 1 mL of DI water under ultrasonication. Then, DMF was removed via dialyzing against DI water to obtain the N$_3$@PGDA7 nanoparticles.

## Physio-chemical characterisation of the POLY-PROTAC, N$_3$@POLY-PROTAC and pretargeted nanoparticles
Hydrodynamic diameter and morphology of all the nanoparticles were investigated by DLS (Zetasizer Nano ZS90, Malvern Instrument, UK) and TEM examination (Talos L120C, USA, 120 kV).

The fluorescence emission profile of the PPa-labelled micelle suspension was examined by microplate reader (Perkin Elmer Enspire, USA), and the fluorescence images were recorded by IVIS imaging system (Xenogen, Alameda, CA).

To demonstrate reduction sensitivity of the POLY-PROTAC nanoparticles, ARV771 release from ARV771 methyl methacrylate (Me-ARV771) was investigated in vitro. Briefly, Me-ARV771 (50 μM) was incubated with 10 mM of DTT (MeOH: H$_2$O =1:1, v/v) for predetermined time duration at 37 °C, the concentration of ARV771 released was examined by HPLC measurement (mobile phase: A: MeOH, C: H$_2$O with 0.1% TFA; flow rate: 1 mL/min; column temperature: 30 °C; UV: 254 nm; elute gradient: 0-10 min, from 5%A to 95% A; 15-20 min, from 95%A to 5%A).

**The PROTAC release profile of the POLY-PROTAC nanoparticle in vitro**

To evaluate ARV771 release profile of the POLY-ARV771 nanoparticle, the suspension of PGD7 nanoparticles was dialysed against 1% (wt) Tween 80-containing buffer solution at 37 °C (MWCO 3500 Da; e.g., pH 6.0 without GSH, pH 6.0 + 10 mM GSH; pH 7.4 without GSH, pH 7.4 + 10 mM, was added in each buffer). ARV771 concentration in the buffer solution was examined by HPLC measurement at the pre-determined time points (e.g., 0.5, 1.0, 2.0, 4.0, 8.0, 12 and 24 h).

**Western blot assay**

To investigate PROTAC-induced protein degradation in vitro, MBA-MB-231 tumour cells were incubated with free PROTACs or POLY-PROTAC nanoparticles at the predetermined condition. The cells were then lysed in RIPA lysis buffer (50 mM Tris-HCl pH 7.4, 150 nM NaCl, 1% NP-40, 0.1% SDS) including 1 mM of phenylmethanesulfonyl fluoride (PMSF). The protein lysate was obtained by centrifugation at 12000 g for 15 min at 4 °C. Then, the lysates were denatured at 100 °C and resolved by SDS-polyacrylamide gel electrophoresis (SDS-PAGE). The gels were blotted onto PVDF membrane (Merck Millipore) and blocked by 5% BSA buffer for 2 h at room temperature. After incubated with the indicated primary antibodies (anti-BRD4 antibody, ab128874, 1:1000; anti-c-Myc antibody, ab32072, 1:1000; anti-caspase-3 antibody, ab 184787, 1:1000; anti-GAPDH antibody, ab8245, 1:1000; anti-β-actin antibody, ab8226, 1:1000, anti-β-tubulin antibody, ab78078, 1:1000) at 4 °C overnight, the membrane was washed several times with TBST buffer, incubated with corresponding secondary antibody (Perox-idase-Conjugated Goat Anti-Rabbit IgG (H + L), CAT:33101ES60, 1:5000 and Peroxidase-Conjugated Goat Anti-Mouse IgG (H + L), CAT:33201ES60, 1:5000) in 5% BSA solution for 1 h at room temperature, washed several times with TBST buffer, and then imaged with Bio-Rad Chemi Doc XRS imaging system.

**Cell proliferation assay**

To investigate the cytotoxicity of the POLY-PROTAC nanoparticles in vitro, $1 \times 10^4$ MDA-MB-231 cells were cultured in 96-well cell culture plate for 24 h, the cells were then treated with free PROTAC or POLY-PROTAC nanoparticles at predetermined condition for 72 h. The cell viability was then examined by CCK-8 assay.

**Cellular uptake and tumour penetration analysis in vitro**

To investigate cellular uptake profile of the POLY-PROTAC nano-particles, MDA-MB-231 cells cultured in 24-well tissue culture plate were incubated with various POLY-PROTAC nanoparticles at the identical PPa concentration of 5.0 μM for the desired time durations (e.g., 2, 4, 8, 12, 24 h, respectively), the intracellular fluorescence intensity were then examined by flow cytometry (BD FACS Fortessa, BD, USA). The intracellular distribution of the POLY-PROTAC nano-particles was investigated by CLSM examination.

3D multicellular spheroid (MCS) of the MDA-MB-231 breast tumour was employed to illustrate the tumour penetration profiles of the POLY-PROTAC nanoparticles in vitro. Briefly, the 3D tumour spheroids were prepared by culturing MDA-MB-231 cells in 48-well tissue culture plate (2000 cells/well) on 1% agarose gel. The 3D tumour spheroids were then incubated with the PGDA7 and PDA7 (MMP-2 pretreated) nanoparticles for 12 h. Nanoparticle distribution inside the tumour spheroids was then investgiated by CLSM examination at the predetermined time points.

**Photoactivity of the POLY-PROTAC nanoparticles in vitro**

To investigate photoactivity of the PPa-modified POLY-PROTAC nanoparticles, 671 nm laser irradiation-induced ROS generation was examined. Briefly, the PGDA suspension with different PPa con-centrations (e.g., 2.5, 5.0, or 10 μM) was incubated in pH 6.0 or pH 7.4 buffer solution for 30 min, and then added with singlet oxygen

sensor green (SOSG, 5.0 μM). The micellar suspension was then irradiated with 671 nm laser at different photodensity (e.g., 100, 200 or 400 mW/cm$^2$) for 1 min, and the fluorescence intensity of SOSG were detected by microplate reader (Perkin Elmer Enspire, USA).

To examine the photoactivity in vitro, MDA-MB-231 cells were culture in 24-well tissue culture plate for 24 h, incubated with the PGDA nanoparticles for 12 h, and then incubated with DCFH-DA (10 μM) for 30 min. The cells were thereafter irradiated with 671 nm laser for 1 min at varied photodensity (e.g., 100, 200, 400 mW/cm$^2$), and the intra-cellular fluorescence intensity of DCF was analysed by flow cytometry. Laser irradiation induced intracellular ROS generation was validated by CLSM examination.

**Biodistribution of the nanoparticles in vivo**

MDA-MB-231 breast tumour model was established by subcutaneously injecting 100 μL of cell suspension ($1 \times 10^7$ cells, DMEM: Matrigel = 1:1, v/v) into the fat pad of the Balb/c nude mice (female, 4 weeks, 18 ~ 20 g). The tumour-bearing mice were randomly grouped when the tumour volume reached ~200 mm$^3$ and intravenously (i.v.) injected with PPa-labelled POLY-PROTAC nanoparticles at an identical PPa dose of 5.0 mg/kg. Then, fluorescence imaging was collected with the IVIS imaging system at predetermined time points (Xenogen, Alameda, CA). The mice were sacrificed at 48 h post-injection, the major organs and tumour tissue were harvested for fluorescence imaging ex-vivo. Afterwards, the tumour tissues were fixed, sectioned and stained with identified antibodies and DAPI, the fluorescence signal of the tumour sections were measured by CLSM examination.

**Bioorthogonal click reaction-enhanced tumour distribution of the POLY-PROTAC nanoparticles in vivo**

To investigate whether in-situ click reaction increased tumour accu-mulation of the POLY-PROTAC nanoparticles, the tumour-bearing nude mice were divided into three groups when the tumour volume reached ~ 200 mm$^3$ ($n = 3$). The DBCO-loaded pretargeted (PED) nanoparticle was i.v. injected at a DBCO dose of 1.0 mg/kg, and the N$_3$@PGDA7 nanoparticle was i.v. injected 2 h post PED administration at an ARV771 dose of 10 mg/kg. The biodistribution of PED and N$_3$@PGDA7 was then examined by IVIS fluorescence imaging systems in vivo.

To evaluate in-situ bioorthogonal click reaction enhanced tumour distribution of PROTAC, the tumour-bearing mice were i.v. injected with the POLY-PROTAC nanoparticles and sacrificed at the pre-determined time points (12, 24, 36, 48 h). The major organs (e.g., heart, liver, spleen, lung and kidney) and tumour tissue were harvested and homogenised. ARV771 was extracted with methanol, examined by HPLC measurement and normalised with tissue mass.

**Anti-tumour study in vivo**

The anti-tumour performance of the POLY-PROTAC nanoparticle was evaluated in MDA-MB-231 orthotropic tumour model in vivo. The female Balb/c nude mice were s.b. injected with 100 μL of cell sus-pension ($1 \times 10^7$ cells, DMEM: Matrigel = 1:1, v/v). The tumour volume reached 100 mm$^3$ at 15-days post inoculation. The tumour-bearing mice were randomly grouped ($n = 5$), and i.v. injected with PBS, ARV771, or PGD7 nanoparticle at an identical ARV771 dose of 10 mg/kg for five cycles at a time interval of 3 days. The tumour volume was monitored for 30-days post treatment.

To investigate the antitumor performance of the bioorthogonal POLY-PROTAC nanoparticles, the tumour-bearing mice were randomly grouped when the tumour volume reached 100 mm$^3$ ($n = 5$), the tumour-bearing mice were then i.v. injected with PBS, ARV771, PGD7, or PED + N$_3$@PGD7 every three days at an identical ARV771 dose of 10 mg/kg, and DBCO dose of 1.0 mg/kg for five cycles (PED was injec-ted 2 h pre- POLY-PROTAC nanoparticle administration).

To investigate the antitumor performance of POLY-PROTAC nanoparticle-performed combinatory BRD4 degradation and PDT, the POLY-PROTAC nanoparticles were labelled with PPa for PDT. The tumour-bearing mice were grouped when the tumour volume reached 100 mm$^3$ ($n = 5$), and then i.v. injected with PBS, ARV771, PED + N$_3$@PGDA, PED + N$_3$@PGD7 or PED + PGDA7 at the identical ARV771 dose of 10.0 mg/kg, PPa dose of 5.0 mg/kg and DBCO dose of 1.0 mg/kg, respectively (PED was i.v. injected 2 h pre-administration of the N$_3$@PGD7 nanoparticles). Thirty-six hours post nanoparticle injection, the mice of the PED + N$_3$@PGDA and PED + PGDA7 groups were treated with 671 nm laser irradiation at photodensity of 400 mW/cm$^2$ for 5 min. The treatment was repeated for five cycles at a time interval of three days. The tumour volume and body weights of the animals were monitored during the whole experimental period. The mice were considered death when the tumour volume reached 1500 mm$^3$ according to the animal ethics of our institute and animal welfare. The tumours and the major organs (e.g., hearts, livers, lungs, spleens and kidneys) of mice were harvested at the end of the antitumor studies, and examined by H&E and TUNEL staining. BRD4 degradation in the tumour tissue was validated by western blot assay of the tumour lysate. The tumour volume was calculated with the formula below:

$$V = L \times W \times W / 2 \; (L, \text{the longest dimension}; W, \text{the shortest dimension}).$$

### Statistical analysis
All data were given as Mean ± SD. The GraphPad Prism software 6.01 was used for the statistical analyses. Two-tailed Student's t-test was used for the statistical comparison between the two groups. One-way analysis of variance (ANOVA) with a Tukey post hoc test was used for statistical comparison among multiple (more than two) groups. Two-sided log-rank (Mantel-Cox) test was used for the statistical comparison of the survival study.

### Reporting summary
Further information on research design is available in the Nature Research Reporting Summary linked to this article.

## Data availability
The authors declare that the data supporting the findings of this study are available within the article, source data, and its Supplementary Information. The source data underlying Figs. 2, 3, 4, 5, 6, supplementary Figs. 12, 32, 33, 34, 35, 40, 41, 42, 43, 44, and western bot are provided with this paper. A reporting summary for this article is available as a Supplementary Information file. Source data are provided with this paper.

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

## Acknowledgements

This work was supported by the National Natural Science Foundation of China (52111530092 and 51873228 to H.J.Y., 22074043 to Z.A.X, 81725008 to H.X.X.), International Cooperation Project of Science and Technology Commission of Shanghai Municipality (20430711800 to H.J.Y.), Science and Technology Commission of Shanghai Municipality (19DZ2251100 to H.X.X.), Shanghai Municipal Health Commission (SHSLCZDZK 03502 and 2019LJ21 to H.X.X.), Scientific Research and Development Fund of Zhongshan Hospital of Fudan University (2022ZSQD07 to H.X.X.), and Open Funds of State Key Laboratory of Drug Research, Shanghai Institute of Materia Medica, CAS (SIMM2105KF-12 to H.J.Y.). The Mass Spectrometry System and the cell sorter BD Influx of the National Facility for Protein Science in Shanghai (NFPS), Shanghai Advanced Research Institute, CAS are gratefully acknowledged. All animal procedures were carried out under the guidelines approved by the Institutional Animal Care and Use Committee (IACUC) of the Shanghai Institute of Materia Medica, CAS.

## Author contributions

H.J.Y. and J.G. conceived the project, J.G., H.J.Y., Z.A.X and H.X.X. designed the study. Q.W.Z. and X.Y.J. synthesised the small molecule PROTACs and their methacrylate derivatives. J.G. and B.H. conducted the click reaction study. J.G. synthesised the POLY-PROTACs and evaluated the protein degradation profiles in vitro and in vivo. J.G., L.Y., and Z.F.Z. conducted the anti-tumour study. T.F.X., X.T.L. and M.Y.Z. provided assistance with PROTAC design. J.G. and H.J.Y. analysed the data and prepared the initial manuscript. T.F.X., Y.H.C., Z.A.X. and H.X.X. revised the manuscript.

## Competing interests

The authors declare no competing interests.
