## [Peer Review File · Nature Communications]

Engineered Bioorthogonal POLY-PROTAC Nanoparticles for Tumour-Specific Protein Degradation and Precise Cancer TherapyREVIEWER COMMENTS

Reviewer #1 (Remarks to the Author):

The manuscript entitled Engineered biorthogonal POLY-PROTAC nanotherapeutics for tumor-specific protein degradation and precise cancer therapy by Jing Gao et al is an interesting manuscript. PROTACs are interesting tools which are largely limited to the preclinical space due to difficulties with in vivo tumor targeting/delivery and specificity. Here, the authors propose a unique approach to this problem using engineered BRD4-targeting POLY-PROTAC nanoparticles given in combination with DBCO-loaded NPs (to improve tumor trapping) and PDT. The data appear high quality and the major conclusions appear supported by the data. This manuscript could be of interest to the general readership of Nature Communications. It should be considered for publications after the authors consider the following revisions:

Comment 1: The Azide-DBCO click reaction is not well described. It is probably unclear to a number of readers why this affects tumor particle distribution. Additional detail may be helpful for a general audience.

Comment 2: Several conclusions are based on representative images of a single replicate with no corresponding quantitative information or information on the number of biological replicates, variance, etc. For example, in Fig 3C, the authors present a CLSM images of PGDA4 and PDA7 particles and conclude that PDGA4 particles demonstrate remarkably higher intracellular uptake compared to PDA7 particles. Qualitatively, this appears true, but for publication in a high impact journal this should be presented with corresponding quantitative data (such as total intracellular integrated density, etc) and appropriate statistical comparisons. Similarly, the Western blot shown in image 4L appears to indicate that ARV771 NPs induced similar levels of cleaved caspase-3 as PGD-7 particles despite having no apparent effect on BRD4 levels. Did both have similar effects on cleaved caspase-3 or was this a replicate anomaly?

Comment 3: Some of the in vivo tumor growth delay and survival comparisons are questionable. The authors indicate that Fig. 6C shows that PED + N2-PGD7 NPs improved survival more than PGD7 alone but assuming they use Kaplan-Meyer analysis to compare survival this seems very unlikely have achieved statistical significance. Similarly, Fig. 6G indicates a statistical difference in tumor volume between PED + PGD7 + laser vs PED +N2-PDG7 + laser groups. However, at most the show very modest separation in tumor volumes at a single time point. What test is being used to compare tumor growth curves and what, specifically, is being compared between groups (the overall curve, tumor volume at a given time point, etc.)? Similarly, in line 310 they indicate that Fig. 4F shows that free ARV771 marginally suppresses tumor growth. It appears extremely unlikely there a statistical difference in tumor volume between ARV771 and PBS treatments when accounting for variance. It would be more accurate (and helpful) for the authors to state that ARV771 particles had no significant effect on tumor growth.

Comment 4: There is a typo in line 394 (enhance should read enhances)

Comment 5: The sentence in line 514 is oddly worded (It remains one priority). Consider rewording.

Comment 6: What is meant by the term tumor-specific ambulation in line 524?

Comment 7: In line 547 the authors indicate that PDT + POLY-PROTAC nps completely eradicated TNBC tumors but survival data in fig. 6H suggests that complete tumor clearance was achieved at most in less than half of subjects in the laser + N2-PDG7 treated animals.

Comment 8: Consider replacing the word ingenious in line 552 with proposed.

Comment 9: A few significant weaknesses should be addressed:

- 1) Initial tumoral delivery of engineered nanoparticles is still dependent upon general EPR effects which are inherently more pronounced in xenografts than de novo or orthotopic tumors.
- 2) PDT is much less commonly used clinically than chemotherapy or radiation. One could easily envision POLY-PROTACs being used to manipulate tumor sensitivity to more commonly utilized anticancer treatments with better specificity than small molecule inhibitors.

Reviewer #2 (Remarks to the Author):

This manuscript by Gao et al. presents a new nanoparticle platform responding to different cues to elicit tumor specific Brd4 protein degradation. This platform exploits published small molecule bivalent degraders of Brd4 and combined them to nanoparticles responding to reduction and to metalloproteinase-2 (MMP2). In addition, through an in situ click reaction the authors manage to trap these nanoparticles in tumors by using a second set of nanoparticles. This makes for a very complex system and sometimes hard to follow story. I think the different nanoparticles would benefit from either being numbered or having easier to understand acronyms. Some acronyms differ from one letter, and it is sometimes hard to keep track of which is the active NP and which one is a control. The paper would benefit from a table, or a list (maybe in the supplementary information) of acronyms and abbreviations used to keep track of the different compounds. The compounds and NP are well characterized, and all the spectra annotated. The results obtained by the authors are significant and offer an interesting use of PROTAC and nanoparticles, never seen before. This complex work capitalizes on a lot of different advances in both the field of nanoparticles and small molecule therapeutics to combine them in one single, sometimes a bit too complex system. I think after clarifying and explaining the system in more details, the paper will gain in clarity and highlight even more these interesting results. The platform developed by the authors is very complex and I am not sure it would be adapted as a real "nanotherapeutic" as the author named their strategy. It is an complex combination of different strategies to increase cytotoxicity and apoptosis, even though it seems a bit convoluted. The in vivo data obtained by the authors is convincing and clearly shows that their strategy works as expected. This new methodology and combination of NP and PROTACs, alongside photodynamic therapy, while not game changing could improve on some of the limitations of the PROTAC technology.

- The figures are very dense and sometimes a bit too small. They would benefit from bigger fonts. Some of the microscopy images would benefit from high resolution images in the supporting information or in a separate file.
- Line 79. More than 2 light induced PROTAC activation have been reported (PhotoPROTACs: 10.1021/acscentsci.9b00713 and PHOTACs: DOI: 10.1126/sciadv.aay5064) and a specific review covering the topic has been published (<https://doi.org/10.3389/fchem.2021.639176>).
- Line 90: PROTAC is misspelled
- Line 94 – 98: since the system developed in this paper is so complex, I think this paragraph introducing the complex could be clearer. Similarly, the figure representing the overall strategy is unclear. I did not understand until later in the paper where the DBCO moiety was coming from. I think it needs to be made clear that the DBCO is located on another nanoparticle and how this nanoparticle allows for retention inside the tumor.
- Line 99: released instead of relieved
- Line 122: the acronym DPA needs to be defined before it is used for the first time.
- Line 123-124 and about DBCO-loaded pretargeted NP: I think the concept of pretargeted NP is not clearly explained in the paper. I would suggest that the authors take time to explain what they are referring to or add clear references directing the readers to previous papers.
- Line 130: the acronym PDT needs to be defined
- Line 131, VHL-based PROTACs: references used previously should be added to more easily identify where these PROTAC structures have been taken from.
- Line 143-145: The importance of the hydroxy group on the VHL ligand has been demonstrated before and several crystal structures are available. The docking of this ligand on VHL was not necessary. The abbreviation HIE for an amino acid is unknown, I suppose the authors mean His115

in the VHL sequence.

- Line 146-148: was the activity of the “caged” PROTACs tested or is there a report of VHL-PROTACs that the authors could cite to prove that addition of a methacrylate abolishes the PROTAC’s activity?
- Line 164: PROTAC molecule instead of PROTAC molecular
- Line 164 – 168: Is there any reference for the MMP-2 cleavable sequence chosen? And has this strategy been used previously for nanoparticles tumor uptake? Line 168, the word “with’ seems to be missing “without GG peptide (PDM), and XXX GG peptide spacer (PGDM)”
- Figure 2 and Figure 3: on some of the western blots, the signal appears to be saturated for β -actin. It may be complicated to quantify the WB using saturated signals (e.g. Figure 3M). Although visible by naked eye, quantification of the WB in the SI would be appreciated.
- Figure 3N: How does the cytotoxicity of the NP compare to the PROTACs alone? It would be good to see if there is any change compared to the active molecule.
- Figure 4H and 4I: due to the size and the resolution of the picture it is tough to judge if the difference observed is due to contrast or a reduction in Brd4 staining.
- Line 339 and Figure 4L: It would have been interesting to see the effect of the nanoparticles without PROTAC on the cleavage of caspase 3. Can the authors comment on the cytotoxicity of their NP without PROTACs?
- Figure 5G and 5I: Where the PED NP tested until 48h like the azide-bearing NP? Why did the authors stop the imaging at 8h while they continued for the other NP? Maybe the remaining images could be added in the SI.
- Line 527: the authors comment on the ability of the NP to carry different amounts of PROTAC small molecule. However, they did not test or show any data on this topic. Therefore, this conclusion although understandable has no data supporting it.

Response to the reviewer

Reviewer 1:

The manuscript entitled Engineered biorthogonal POLY-PROTAC nanotherapeutics for tumor-specific protein degradation and precise cancer therapy by Jing Gao et al is an interesting manuscript. PROTACs are interesting tools which are largely limited to the preclinical space due to difficulties with in vivo tumor targeting/delivery and specificity. Here, the authors propose a unique approach to this problem using engineered BRD4-targeting POLY-PROTAC nanoparticles given in combination with DBCO-loaded NPs (to improve tumor trapping) and PDT. The data appear high quality and the major conclusions appear supported by the data. This manuscript could be of interest to the general readership of Nature Communications. It should be considered for publications after the authors consider the following revisions:

Response: We appreciate the insightful comments of the reviewer.

1: The Azide-DBCO click reaction is not well described. It is probably unclear to a number of readers why this affects tumor particle distribution. Additional detail may be helpful for a general audience.

Response: To improve readability of the manuscript, we revised the sentences in Line 95-105 as below:

“To enforce the tumour specificity of POLY-PROTAC, a pretargeted NP was subsequently engineered for the tumour-specific delivery of dibenzocyclooctyne (DBCO) groups. This pretargeted NP can be activated in the tumour mass due to the acidic tumour microenvironment (i.e., pH = 6.5~6.8), which exposes the DBCO group in the tumour to cross-link with the azide groups of the secondary POLY-PROTAC NPs via an in situ click reaction³¹; this occurs via copper-free catalysis in the biological milieu³². The azide-modified POLY-PROTAC NPs thus become entrapped and retained in the tumour mass by suppressing blood clearance of the NPs. The POLY-PROTAC NPs can subsequently be released into the extracellular matrix (ECM) of the tumours via metalloproteinase-2 (MMP-2)-mediated cleavage of the PEG corona.”

32. Wang, H., Mooney, D.J. Metabolic glycan labelling for cancer-targeted therapy. *Nat Chem.* **12**,

2: Several conclusions are based on representative images of a single replicate with no corresponding quantitative information or information on the number of biological replicates, variance, etc. For example, in Fig 3C, the authors present a CLSM images of PGDA7 and PDA7 particles and conclude that PDGA4 particles demonstrate remarkably higher intracellular uptake compared to PDA7 particles. Qualitatively, this appears true, but for publication in a high impact journal this should be presented with corresponding quantitative data (such as total intracellular integrated density, etc) and appropriate statistical comparisons. Similarly, the Western blot shown in image 4L appears to indicate that ARV771 NPs induced similar levels of cleaved caspase-3 as PGD-7 particles despite having no apparent effect on BRD4 levels. Did both have similar effects on cleaved caspase-3 or was this a replicate anomaly?

Response: We added the semiquantitative data of intracellular fluorescence density in Supplementary Figure 32b of the revised manuscript. PGDA7 NPs displayed ~ 3.0-fold higher intracellular fluorescence intensity in MDA-MB-231 cells than the PDA7.

Fig. S32. (a) CLSM examination of intracellular distribution of the PPa-labeled POLY-PROTAC nanoparticles in MDA-MB-231 tumor cells *in vitro*, and (b) Integrated intracellular fluorescence intensity of POLY-PROTAC nanoparticles-treated MDA-MB-231 tumor cells upon 12 h incubation (n = 3, *** p < 0.001);

We also integrated the gray intensity of the western blot image in Fig. 4h with triple replicates. The data was provided as Supplementary Figure 34b of the revised manuscript. Semi-quantitative integration of the western blot bands revealed that the PGD7 POLY-PROTAC NPs 1.5-fold more efficiently induced caspase-3 activation than free ARV771.

Fig. S34b. Semi-quantitation of POLY-PROTAC nanoparticles-induced caspase-3 activation. Western blot assay of PGD7 POLY-PROTAC-induced caspase-3 activation in MDA-MB-231 tumor cells *in vivo* (the tumors were harvested at the second day post five-cycles treatments) (n = 3).

3: Some of the *in vivo* tumor growth delay and survival comparisons are questionable. The authors indicate that Fig. 6C shows that PED + N₃@PGD7 NPs improved survival more than PGD7 alone but assuming they use Kaplan-Meyer analysis to compare survival this seems very unlikely have achieved statistical significance. Similarly, Fig. 6G indicates a statistical difference in tumor volume between PED + PGD7 + laser vs

PED +N2-PDG7 + laser groups. However, at most they show very modest separation in tumor volumes at a single time point. What test is being used to compare tumor growth curves and what, specifically, is being compared between groups (the overall curve, tumor volume at a given time point, etc.)? Similarly, in line 310 they indicate that Fig. 4F shows that free ARV771 marginally suppresses tumor growth. It appears extremely unlikely there is a statistical difference in tumor volume between ARV771 and PBS treatments when accounting for variance. It would be more accurate (and helpful) for the authors to state that ARV771 particles had no significant effect on tumor growth.

Response: We appreciate the critical comments of the reviewer.

Exactly, the t-test revealed insignificant difference ($p = 0.1218$) between the survival curves of the PED + N₃@PGD7 NPs and PGD7 groups due to tumor relapse at the late stage of the antitumor study, although PED + N₃@PGD7 significantly delayed tumor growth than the PGD7 groups in 27-days post first nanoparticle injection (* $p = 0.0492$). To describe the data more accurately, we revised the sentences at Line 444-449: “Compared to free ARV771 or N₃@PGD7 NPs, the combination of the PED pretargeted and N₃@PGD7 NPs more efficiently delayed ~70% of MDA-MB-231 tumor growth due to increased ARV771 distribution in the tumor tissue (Fig. 6b). However, PED + N₃@PGD7 insignificantly elongated the survival of the tumor-bearing mice compared to N₃@PGD7 NPs alone due to tumor relapse and animal death at the end of the antitumor study (Fig. 6c).”

Furthermore, as pointed out by the reviewer, there is no statistical difference between the tumor growth rate of PBS and free ARV771 groups with a P value of 0.3737 (Fig. 4f).

We corrected the description in Line 310-312:

“Free ARV771 negligibly affected tumor growth compared to PBS control.”

In the antitumor study shown in Fig. 6g, we did observe a statistical difference between the growth curves of the PED + PGDA7 + Laser (without click reaction) and the PED + N₃@PGDA7 + Laser groups ($p < 0.0001$, t-test). The tumor growth plots of these two groups are separated when they are zoomed in as below.

Supplementary Figure 1 (for review only). Averaged and individual tumor growth curves after treatment with PED + PGD7 + Laser and PED + N₃@PGDA7 + Laser, respectively (n = 5, irradiation condition: 400 mW/cm², 5.0 min; **** p < 0.0001, t test).

4: There is a typo in line 394 (enhance should read enhances)

Response: “enhance” was corrected as “enhances”.

5: The sentence in line 514 is oddly worded (It remains one priority). Consider rewording.

Response: The sentence was revised as “It remains challengeable to develop novel PROTACs for tumor-specific protein degradation while minimize the on-target but off-tumor adverse effects.”

6: What is meant by the term tumor-specific ambulation in line 524?

Response: The term was corrected as “tumor-specific accumulation”.

7: In line 547 the authors indicate that PDT + POLY-PROTAC NPs completely eradicated TNBC tumors but survival data in Fig. 6H suggests that complete tumor clearance was achieved at most in less than half of subjects in the laser + N₃@PDG7-treated animals.

Response: Exactly, the sentence in Line 547 over-estimated the therapeutic efficacy of PDT + POLY-PROTAC NPs, Fig. 6g displayed that the combination of the bioorthogonal NPs (PED + N₃@PGDA7) and PDT dramatically regressed 90% of tumor growth in the experimental period, with 40% of the tumor-bearing mice survived in 90-days post treatment (Fig. 6h).

To avoid confusion, we revised the sentence in Line 552-553: “which suppressed 95% of MDA-MB-231 TNBC tumor growth”.

8: Consider replacing the word ingenious in line 552 with proposed.

Response: Done.

9. Initial tumoral delivery of engineered nanoparticles is still dependent upon general EPR effects which are inherently more pronounced in xenografts than de novo or orthotopic tumors.

Response: Exactly, tumor distribution of the POLY-PROTAC nanoparticle alone in our MDA-MB-231 xenograft tumor model is dependent on the EPR effect. Furthermore, to improve tumor accumulation and retention of the POLY-PROTAC nanoparticles, in this study we engineered tumor acidity-activatable pretargeted nanoparticles for tumor-specific delivery of DBCO group, which can crosslink and entrap the azide-labeled POLY-PROTAC nanoparticles inside the tumor mass via in-situ click reaction. The EPR effect⁴¹, and the acidic tumor microenvironment (i.e., pH = 6.5-6.8)³¹ had both been well documented in both the xenografts and orthotopic tumor models. Therefore, we presumed that the POLY-PROTAC nanoparticles could specifically distributed in the transgenic or orthotopic tumor models, however, which remains to be exploited in future investigation.

41. Matsumura, Y. Preclinical and clinical studies of NK012, an SN-38-incorporating polymeric micelles, which is designed based on EPR effect. *Adv Drug Deliv Rev.* **63**, 184-192 (2011).

31. Huang, G., Zhao, T., Wang, C., Nham, K., Xiong, Y., Gao, X., Wang, Y., Hao, G., Ge, W.P., Sun, X., Sumer, B.D., Gao, J. PET imaging of occult tumours by temporal integration of tumour-acidosis signals from pH-sensitive ⁶⁴Cu-labelled polymers. *Nat Biomed Eng.* **4**, 314-324 (2020).

10. PDT is much less commonly used clinically than chemotherapy or radiation. One could easily envision POLY-PROTACs being used to manipulate tumor sensitivity to more commonly utilized anticancer treatments with better specificity than small molecule inhibitors.

Response: Exactly, PDT is currently exploited in the (pre)clinical investigations. Apart from PDT, the POLY-PROTAC could be further engineered for combination with chemotherapy and radiotherapy. Both of these combinations are under investigation in our laboratory. To increase the impact of our study, we revised the discussion in Line 537-542: “Moreover, other kind of stimuli-activatable chemical bonds (e.g., thioketone, selenic or boric acid bond) can be utilized to achieve spatial-temporally PROTAC activation. Apart from PDT, the POLY-PROTAC nanoplatform can be combined with other kinds of therapy modalities (e.g., radiotherapy or chemotherapy), which are both under investigation our laboratory for potentiated the antitumor performance.”

Reviewer #2

This manuscript by Gao et al. presents a new nanoparticle platform responding to different cues to elicit tumor specific Brd4 protein degradation. This platform exploits published small molecule bivalent degraders of Brd4 and combined them to nanoparticles responding to reduction and to metalloproteinase-2 (MMP2). In addition, through an in situ click reaction the authors manage to trap these nanoparticles in tumors by using a second set of nanoparticles. This makes for a very complex system and sometimes hard to follow story. I think the different nanoparticles would benefit from either being numbered or having easier to understand acronyms. Some acronyms differ from one letter, and it is sometimes hard to keep track of which is the active NP and which one is a control. The paper would benefit from a table, or a list (maybe in the supplementary information) of acronyms and abbreviations used to keep track of the different compounds.

The compounds and NP are well characterized, and all the spectra annotated. The results obtained by the authors are significant and offer an interesting use of PROTAC and nanoparticles, never seen before. This complex work capitalizes on a lot of different advances in both the field of nanoparticles and small molecule therapeutics to combine them in one single, sometimes a bit too complex system. I think after clarifying and explaining the system in more details, the paper will gain in clarity and highlight even more these interesting results. The platform developed by the authors is very complex and I am not sure it would be adapted as a real “nanotherapeutic” as the author named their strategy. It is a complex combination of different strategies to increase cytotoxicity and apoptosis, even though it seems a bit convoluted. The in vivo data obtained by the authors is convincing and clearly shows that their strategy works as expected. This new methodology and combination of NP and PROTACs, alongside photodynamic therapy could improve on some of the limitations of the PROTAC technology.

Response: We sincerely appreciate the insightful and critical comments of the reviewer. As suggested by the reviewer, we added a Cartoon as Supplementary Figure 31 to illustrate the compositions and acronyms of the nanoparticles investigated in this study.

We also replaced ‘Nanotherapeutics’ with ‘Nanoparticles’ in the manuscript title.

Supplementary Fig. S31. Cartoon illustration of nanoparticle compositions and acronyms investigated throughout our study.

In this study, we rationally designed a polymeric PROTAC (POLY-PROTAC) nanoparticle for tumor-specific protein degradation and improved cancer therapy. The POLY-PROTACs were engineered by covalently grafting small molecular PROTACs onto the backbone of an amphiphilic diblock copolymer via a disulfide bond. The resultant POLY-PROTACs can be self-assembled into micellar nanoparticles and displayed tumor-targeted BRD4 degradation performance (Fig. 4a-d).

To increase intratumoral accumulation and retention of the POLY-PROTAC NPs, we functionalized the POLY-PROTAC NPs with azide group for in-situ click reaction-amplified PROTAC delivery and protein degradation in the tumor tissue *in vivo*. For this purpose, a DBCO-loaded PED pretargeted nanoparticle was developed for delivering DBCO group into the tumor extracellular matrix and trapping the azide-modified POLY-PROTAC nanoparticles via in-situ click reaction. The bioorthogonal

click reaction strategy remarkably increased tumor distribution of the POLY-PROTAC nanoparticles (Fig. 5i-m).

We further combined the clickable POLY-PROTAC nanoparticles with PDT for prompting their antitumor performance since PDT and BRD4 degradation synergistically induced apoptosis of the tumor cells (Fig. 6d, e). Overall, this study demonstrated a novel strategy for tumor-targeted PROTAC delivery and improved cancer therapy via PROTAC technology.

1. The figures are very dense and sometimes a bit too small. They would benefit from bigger fonts. Some of the microscopy images would benefit from high resolution images in the supporting information or in a separate file.

Response: As suggested by the reviewer, we amplified the fonts in all the Figures in the revised manuscript. We also uploaded high resolution images for all the CLSM and western blot data by following the guideline of Nature Communications.

2. Line 79. More than 2 light induced PROTAC activation have been reported (PhotoPROTACs: 10.1021/acscentsci.9b00713 and PHOTACs: DOI: 10.1126/sciadv.aay5064) and a specific review covering the topic has been published (<https://doi.org/10.3389/fchem.2021.639176>).

Response: We cited above three publications as new references 25-27 in Line 81.

25. Pfaff, P., Samarasinghe, K.T.G., Crews, C.M., Carreira, E.M. Reversible Spatiotemporal Control of Induced Protein Degradation by Bistable PhotoPROTACs. *ACS Cent Sci.* **5**, 1682-1690 (2019).

26. Reynders, M., Matsuura, B.S., Bérouti, M., Simoneschi, D., Marzio, A., Pagano, M., Trauner, D. PHOTACs enable optical control of protein degradation. *Sci Adv.* **6**, eaay5064 (2020).

27. Zeng, S., Zhang, H., Shen, Z., Huang, W. Photopharmacology of Proteolysis-Targeting Chimeras: A New Frontier for Drug Discovery. *Front Chem.* **9**, 639176 (2021).

3. Line 90: PROTAC is misspelled

Response: Corrected.

4. Line 94 – 98: since the system developed in this paper is so complex, I think this paragraph introducing the complex could be clearer. Similarly, the figure representing the overall strategy is unclear. I did not understand until later in the paper where the DBCO moiety was coming from. I think it needs to be made clear that the DBCO is located on another nanoparticle and how this nanoparticle allows for retention inside the tumor.

Response: To improve readability of the manuscript, we added more explanation about the DBCO-azide click reaction at Line 95-105:

“To improve the tumour specificity of POLY-PROTAC, a pretargeted NP was subsequently engineered for the tumour-specific delivery of dibenzocyclooctyne (DBCO) groups. This pretargeted NP can be activated in the tumour mass due to the acidic tumour microenvironment (i.e., pH = 6.5~6.8), which exposes the DBCO group in the tumour to cross-link with the azide groups of the secondary POLY-PROTAC NPs via an in situ click reaction³¹; this occurs via copper-free catalysis in the biological milieu³². The azide-modified POLY-PROTAC NPs thus become entrapped and retained in the tumour mass by suppressing blood clearance of the NPs. The POLY-PROTAC NPs can subsequently be released into the extracellular matrix (ECM) of the tumours via metalloproteinase-2 (MMP-2)-mediated cleavage of the PEG corona.”

32. Wang, H., Mooney, D.J. Metabolic glycan labelling for cancer-targeted therapy. *Nat Chem.* **12**, 1102-1114 (2020).

5. Line 99: released instead of relieved

Response: Corrected.

6. Line 122: the acronym DPA needs to be defined before it is used for the first time.

Response: DPA was identified as “2-(disopropylamino) ethyl methacrylate (DPA)” in line 93 where it appeared for the first time. We also provided full name of PEG and GSH at Line 93 and Line 107, respectively.

7. Line 123-124 and about DBCO-loaded pretargeted NP: I think the concept of pretargeted NP is not clearly explained in the paper. I would suggest that the authors take time to explain what they are referring to or add clear references directing the readers to previous papers.

Response: We appreciate the critical suggestion of the reviewer.

To provide a clear description of the pretargeted nanoparticles, we revised the sentences in Line 95-100: “To improve tumor specificity of the POLY-PROTAC, a pretargeted NP was subsequently engineered for tumor-specific delivery of dibenzocyclooctyne (DBCO) groups. The pretargeted NP can be activated in the tumor mass by targeting the acidic tumor microenvironment (i.e., pH = 6.5 ~ 6.8), and thus expose the DBCO group in the tumor for reacting with azide groups of the secondary POLY-PROTAC NPs via in-situ click reaction.”

8. Line 130: the acronym PDT needs to be defined

Response: The acronym PDT has been defined as “photodynamic therapy (PDT)” when it first appeared at Line 110.

9. Line 131, VHL-based PROTACs: references used previously should be added to more easily identify where these PROTAC structures have been taken from.

Response: According to the suggestion of the reviewer, we cited two references about VHL synthesis as new references 34-35 in the revised manuscript at Line 134.

34. Raina, K., Lu, J., Qian, Y., Altieri, M., Gordon, D., Rossi, A.M., Wang, J., Chen, X., Dong, H., Siu, K., Winkler, J.D., Crew, A.P., Crews, C.M., Coleman, K.G. PROTAC-induced BET protein degradation as a therapy for castration-resistant prostate cancer. *Proc Natl Acad Sci USA*. **113**, 7124-7129 (2016).

35. Zengerle, M., Chan, K.H., Ciulli, A. Selective Small Molecule Induced Degradation of the BET Bromodomain Protein BRD4. *ACS Chem Biol*. **10**, 1770-1777 (2015).

10. Line 143-145: The importance of the hydroxyl group on the VHL ligand has been demonstrated before and several crystal structures are available. The docking of this

ligand on VHL was not necessary. The abbreviation HIE for an amino acid is unknown, I suppose the authors mean His115 in the VHL sequence.

Response: Exactly, molecular docking and crystal structure of the VHL ligand and VHL protein has been reported previously, we deleted Figure 2e in the revised manuscript, and cited one literature as new reference 36 demonstrating importance of the hydroxyl group on the VHL ligand at Line 146.

36. Naro, Y., Darrah, K., Deiters, A. Optical Control of Small Molecule-Induced Protein Degradation. *J Am Chem Soc.* **142**, 2193-2197 (2020).

The abbreviation “HIE” is corrected as “HIS”. We further revised the sentence in line 144-146 as: “The hydroxyl group on the VHL ligand plays crucial role for VHL protein binding by forming hydrogen-bond with HIS-115 and SER-111 in the binding pocket of VHL protein.”

11. Line 146-148: was the activity of the “caged” PROTACs tested or is there a report of VHL-PROTACs that the authors could cite to prove that addition of a methacrylate abolishes the PROTAC’s activity?

Response: Exactly, the addition of a methacrylate completely abolishes the PROTAC’s activity as displayed in Fig. 3k. Here a reduction-inert ethylene glycol spacer was employed to synthesize the GSH-non-responsive methacrylate of ARV771 (namely Me-O-ARV771) (Supplementary Scheme 6).

Fig. 3k Western blot assay of BRD4 expression in the PGDO7 NP-treated MDA-MB-231 cells *in vitro*.

Supplementary Scheme S6. Synthesis of GSH-insensitive ARV771 methacrylate (Me-O-ARV771).

12. Line 164: PROTAC molecule instead of PROTAC molecular

Response: Corrected.

13. Line 164 – 168: Is there any reference for the MMP-2 cleavable sequence chosen? And has this strategy been used previously for nanoparticles tumor uptake? Line 168, the word “with” seems to be missing “without GG peptide (PDM), and XXX GG peptide spacer (PGDM)”

Response: We cited one of our previous publications as new reference 29 to support the selection of GPLGLAG(GG) peptide as the MMP-2 cleavable sequence for increased tumor accumulation and cellular uptake of the nanoparticles.

Furthermore, “with” was added at Line 167: “with GG peptide spacer (PGDM)”

29. Zhou, F., Gao, J., Tang, Y., Zou, Z., Jiao, S., Zhou, Z., Xu, H., Xu, Z.P., Yu, H., Xu, Z.

Engineering Chameleon Prodrug Nanovesicles to Increase Antigen Presentation and Inhibit PD-L1 Expression for Circumventing Immune Resistance of Cancer. *Adv Mater.* 33, 2102668 (2021).

14. Figure 2 and Figure 3: on some of the western blots, the signal appears to be saturated for β -actin. It may be complicated to quantify the WB using saturated signals (e.g. Figure 3M). Although visible by naked eye, quantification of the WB in the SI would be appreciated.

Response: As suggested by the reviewer, we adjusted the saturation degree of the western blot images in Fig. 2b, Fig. 3g-k and Fig. 3m in the revised manuscript. We also provided the quantification data in Supplementary Fig. S12 and Fig. S33.

15. Figure 3N: How does the cytotoxicity of the NP compare to the PROTACs alone?

It would be good to see if there is any change compared to the active molecule.

Response: Fig. 2d and Fig. 3n displayed that the MMP-2-liable POLY-PROTAC NPs are of comparable half inhibition concentrations as that of the parental molecule (e.g., PGDM (0.27 μm) Vs. MZ1 (0.55 μm) and PGD7 (0.25 μm) Vs. ARV771 (0.17 μm)). As suggested by the reviewer, we added a comparison between the MMP-2-liable POLY-PROTAC NPs and the active molecules at Line 256-259:

“Notably, the MMP-2-liable POLY-PROTAC NPs displayed half inhibitory concentrations comparable to that of the parental molecule (Fig. 2d), suggesting that the PROTAC payload can be readily released inside the tumour cells via GSH-mediated cleavage of the disulfide bond.”

16. Figure 4H and 4I: due to the size and the resolution of the picture it is tough to judge if the difference observed is due to contrast or a reduction in Brd4 staining.

Response: The amplified BRD4 immune histochemical staining images in Fig. 4i clearly displayed downregulation of BRD4 protein in the PGD7 NP group compared to the PBS and ARV771 groups.

Fig. 4j. H&E staining of the tumor sections at the end of antitumor study (scale bar = 100 μm); **Fig. 4k.** IHC staining of BRD4 expression in the tumor sections (scale bar = 100 μm).

17. Line 339 and Figure 4L: It would have been interesting to see the effect of the nanoparticles without PROTAC on the cleavage of caspase 3. Can the authors comment

on the cytotoxicity of their NP without PROTACs?

Response: Fig. 4i (original Fig. 4l) displayed that PGD7 NP remarkably activated Caspase-3 in MDA-MB-231 tumor cells as a function of ARV771 concentration (Supplementary Figure S34c).

Fig. 4i. Western blot assay of PGD7 POLY-PROTAC-induced BRD4 degradation and caspase-3 activation in MDA-MB-231 tumor cells *in vitro* as a function of ARV771 concentration.

To elucidate the mechanism of PGD7-induced caspase-3 activation, western blot assay was performed to investigate the influence of free ARV771 on cleaved caspase-3 expression *in vitro*. Supplementary Figure S34d&e revealed that free ARV771 cleaved caspase-3 in a concentration-dependent manner, and reached plateau at a ARV771 feeding concentration of 1.0 μ M, validating ARV771 induced apoptosis of the tumor cells through the caspase pathway.

Furthermore, CCK-8 assay displayed that ARV771-free PGD7 NPs induced negligible cytotoxicity in MDA-MB-231 tumor cells *in vitro* at a concentration up to 20 μ g/mL (Supplementary Fig. S41). Therefore, we concluded that caspase-3 activation property of the PGD7 NPs could be attributed to ARV771 instead of the ARV771-free NPs.

Fig. S34. **a** Semi-quantitation of western blot band showed in Fig. 4h for BRD4 expression in MDA-MB-231 tumor examined at the end of the anti-tumor study *in vivo*; **b** Semi-quantitation of western blot band showed in Fig. 4h for caspase-3 activation in the MDA-MB-231 tumor *in vivo* upon ARV771 or POLY-PROTAC NPs treatment (the tumors were harvested at the second day post five-cycles treatments, n = 3); **c** Semi-quantitation of the western blot band in Fig. 4i for PGD7 NP-induced caspase-3 activation in MDA-MB-231 tumor cells *in vitro*; **d** Western blot assay, and **e** Semi-quantitation of the western blot band of ARV771-induced caspase-3 activation in MDA-MB-231 cells *in vitro*. The cells were treated with ARV771 for 24 h and then examined by western blot assay (n = 3).

Fig. S41. Cytotoxicity assay of ARV771-free PGD7 NPs in MDA-MB-231 tumor cells *in vitro*. The cells were incubated with the NPs for 72 h, and examined by CCK-8 assay.

18. Figure 5G and 5I: Where the PED NP tested until 48h like the azide-bearing NP? Why did the authors stop the imaging at 8h while they continued for the other NP? Maybe the remaining images could be added in the SI.

Response: Fluorescence imaging *in vivo* is performed to validate extracellular tumor acidity-triggered activation of the PED nanoparticles. Fig. 5g displayed that 2 h post i.v. injection, the PED NPs were activated in the acidic tumor microenvironment to generate fluorescence signal. CLSM examination of the tumor sections displayed that the PED pretargeted NPs colocalized well with the cell membrane (labeled with wheat germ agglutinin, WGA) 2~4 h post-injection, validating that the PED NPs dissociated and exposed the DBCO groups in the extracellular matrix (ECM) of the tumor tissue (Fig. 5h). Intratumoral dissociation of the PED NPs could facilitate click reaction between the DBCO and azide groups presenting on the surface of the POLY-PROTAC NPs in the ECM. In contrast, the fluorescence signal of the PED NPs appeared inside the tumor cells when examined 8 h post i.v. injection, suggesting cellular uptake of the PED NPs. To facilitate in-situ click reaction between the PED pretargeted NPs and the PROTAC-loaded N₃@PGD7 secondary NPs, we therefore injected the N₃@PGD7 NPs 2h post i.v. injection of the PED ones and monitored tumor distribution of the PED NPs up to 8 h postinjection.

19. Line 527: the authors comment on the ability of the NP to carry different amounts of PROTAC small molecule. However, they did not test or show any data on this topic. Therefore, this conclusion although understandable has no data supporting it.

Response: Exactly, the PROTAC loading ratio of the POLY-PROTAC nanoparticles can be readily tuned by adjusting the polymerization degree of PROTAC in the POLY-PROTAC copolymers. We delete the sentence at Line 527 to avoid over statement since we did not demonstrate the ability of the POLY-PROTAC NPs to adjust PROTAC loading ratios in this study.

REVIEWERS' COMMENTS

Reviewer #1 (Remarks to the Author):

The manuscript "Engineered Bioorthogonal POLY-PROTAC Nanoparticles for Tumour-Specific Protein Degradation and Precise Cancer Therapy" proposes an interesting and novel approach to improving PROTAC nanovectors. The study is well done and in my previous review I requested several points of clarification regarding data interpretation and analysis (along with a number of minor comments related mostly to document presentation). In the revisions and author response document, the authors have adequately addressed all of my previous concerns. This manuscript will be of interest to the readership of Nature Communications.

Reviewer #2 (Remarks to the Author):

The authors addressed most if not all the concerns of both reviewers and produced a much clearer article this time. With the additional figures, analysis and corrections in the main text, I fully support the publication of this article in Nature Communication.